# Hypoxia-induced inhibin promotes tumor growth and vascular permeability in ovarian cancers

Ben Horst[1,2], Shrikant Pradhan[2,8], Roohi Chaudhary [3,8], Eduardo Listik[1], Liz Quintero-Macias[1], Alex Seok Choi[1], Michael Southard[2], Yingmiao Liu[4], Regina Whitaker[5], Nadine Hempel[6], Andrew Berchuck[5], Andrew B. Nixon[4], Nam Y. Lee[7], Yoav I. Henis [3] & Karthikeyan Mythreye [1,2✉]

Hypoxia, a driver of tumor growth and metastasis, regulates angiogenic pathways that are targets for vessel normalization and ovarian cancer management. However, toxicities and resistance to anti-angiogenics can limit their use making identification of new targets vital. Inhibin, a heteromeric TGFβ ligand, is a contextual regulator of tumor progression acting as an early tumor suppressor, yet also an established biomarker for ovarian cancers. Here, we find that hypoxia increases inhibin levels in ovarian cancer cell lines, xenograft tumors, and patients. Inhibin is regulated primarily through HIF-1, shifting the balance under hypoxia from activins to inhibins. Hypoxia regulated inhibin promotes tumor growth, endothelial cell invasion and permeability. Targeting inhibin in vivo through knockdown and anti-inhibin strategies robustly reduces permeability in vivo and alters the balance of pro and anti-angiogenic mechanisms resulting in vascular normalization. Mechanistically, inhibin regulates permeability by increasing VE-cadherin internalization via ACVRL1 and CD105, a receptor complex that we find to be stabilized directly by inhibin. Our findings demonstrate direct roles for inhibins in vascular normalization via TGF-β receptors providing new insights into the therapeutic significance of inhibins as a strategy to normalize the tumor vasculature in ovarian cancer.

[1] Department of Pathology and O'Neal Comprehensive Cancer Center, Heersink School of Medicine, University of Alabama at Birmingham, Birmingham, AL 35233, USA. [2] Department of Chemistry and Biochemistry, University of South Carolina, Columbia, SC 29208, USA. [3] School of Neurobiology, Biochemistry and Biophysics, George S. Wise Faculty of Life Sciences, Tel Aviv University, Tel Aviv 6997801, Israel. [4] Department of Medicine and Duke Cancer Institute, Duke University Medical Center, Durham, NC 27710, USA. [5] Department of Obstetrics and Gynecology, Duke University Medical Center, Durham NC, Duke University, Durham, NC 27710, USA. [6] Department of Medicine, Division of Hematology/Oncology, University of Pittsburgh, Pittsburgh, PA 15213, USA. [7] Division of Pharmacology, Chemistry and Biochemistry, College of Medicine, University of Arizona, Tucson, AZ 85721, USA. [8]These authors contributed equally: Shrikant Pradhan, Roohi Chaudhary. ✉email: mythreye@uab.edu

Changes in angiogenesis are associated with metastasis in most cancers, including ovarian cancers, with significant impact on tumor progression and ascites development in advanced disease[1,2]. As such, anti-angiogenic therapies have had significant impact in the management of ovarian cancers[3]. However, their effectiveness can be frequently limited due in part to toxicities and acquired resistance, leading to challenges with long term use and marginal improvements in overall survival[3]. Discovery of new and safer angiogenic targets is thus critical.

TGFβ family members, particularly BMP9 and TGFβ, are the most examined regulators of angiogenesis but have not been effective as targets for angiogenic therapy due to their pleiotropic functions in cancer and normal physiology[4,5]. Similar to TGFβ and BMP9, activins' have controversial and context dependent roles in angiogenesis. Specifically activin A has been shown to increase VEGF induced angiogenesis in some instances[6] and in others has been demonstrated to inhibit angiogenesis[7]. Inhibins' are a distinct and unique member of the TGFβ family as the only endocrine hormone and a functional heterodimer of an alpha (α) subunit (INHA) and a beta (β) activin subunit (INHBA or INHBB) forming either inhibin A or inhibin B respectively[8]. Inhibins' are distinct from activins which are comprised of dimers of either beta subunit[8]. Inhibinα is synthesized as a pro-peptide with a pro-domain, αN region, and αC region. The pro-domain and αN region can be cleaved to produce the mature Inhibinα subunit comprising the αC region. Physiological Inhibinα production by the sertoli cells of the testes, granulosa cells of the ovary, and the adrenal and pituitary glands[9] is regulated primarily by follicle stimulating hormone (FSH) and luteinizing hormone (LH)[10,11] via a cAMP-PKA (cyclic adenosine monophosphate-protein kinase A) pathway resulting in cAMP response element binding (CREB) to the cAMP response element (CRE) on the INHA promoter[12].

While inhibin levels (inhibin A and B) cycle across the lifespan of healthy females and dramatically decrease at the onset of menopause[13], elevated inhibinα levels are found in ovarian, gastric, hepatocellular, and prostate cancers[14–17]. Total inhibin protein levels comprising free inhibinα, inhibin A and inhibin B are also an established diagnostic marker alone and/or in combination with CA125, for ovarian cancers[18] and have been proposed as a potential tumor specific target for therapy[8,15,17,19,20]. Inhibinα levels are also predictive of survival in multiple cancer types with gene signatures that correlate with INHA expression, providing a highly accurate prognostic model for predicting patient outcomes[20]. However, the mechanism of inhibin expression in cancers have not been delineated.

Hypoxia is a key mediator of angiogenic responses, regulating pro and anti-angiogenic genes impacting tumor growth, metastasis, and immune evasion[21] and is driven by the hypoxia inducible factor (HIF) family of transcription factors. Hypoxia induced changes, specifically in tumors, are characterized by inefficient oxygen delivery, leading to leaky vessels, and altered permeability, build-up of fluid and ascites in ovarian cancer, and metastasis by facilitating intra/extravasation of tumor cells[21,22]. We previously reported decreased ascites accumulation in mice bearing tumor cells with INHA knockdown[19], indicating a potential role for inhibin in regulating metastasis and vascular functions, a key contributing factor to ascites accumulation. Moreover, inhibin secreted by tumor cells induces angiogenesis via SMAD1/5 signaling in endothelial cells in a paracrine manner dependent on the type III TGFβ receptor endoglin/CD105 and the type I TGFβ receptor ALK1[19].

To precisely delineate inhibin's significance in cancer and mechanism of action, we now determine the impact of hypoxia, a key mediator of the angiogenic and metastatic response in cancer[21], and the contribution of inhibin to the hypoxia adaptive response. We discover that hypoxia in ovarian xenograft tumors, cancer cells, and patient samples leads to an increase in inhibin synthesis in a hypoxia inducible factor (HIF) dependent manner. We find that hypoxia induced tumor growth and vascular permeability in vivo is driven by inhibin. Moreover, intervention using an antibody based therapeutic strategy to inhibin can suppress hypoxia driven tumor biology. Mechanistically, inhibin promotes vascular permeability via endoglin and ALK1. Notably, we also define, using sensitive biophysical methods, the nature and stability of the endoglin and ALK1 interaction at the cell surface in response to inhibin. Our findings not just strongly implicate inhibins as part of the hypoxia adaptive response, but also suggest anti-inhibins' as an alternative or companion to current anti-angiogenic therapies that may otherwise not be well tolerated.

## Results

**Expression and secretion of inhibin is regulated by hypoxia in ovarian cancer cell lines**. We and others have previously demonstrated increased expression of inhibinα mRNA and protein in a broad spectrum of cancers leading to increased angiogenesis in vitro and in vivo impacting metastasis[17,19,23]. Based on the potential role of inhibins' in cancer angiogenesis, we tested the impact of hypoxia, a key regulator of angiogenesis, on INHA expression. The high grade serous ovarian cancer cell lines HEY and OV90 cells were exposed to varying levels of oxygen (control tissue culture conditions (20%), 10%, 5%, 2.5%, 1%, and 0.2% $O_2$) for 24 h to evaluate INHA expression and VEGFA expression (as a positive control[24]) by semi-quantitative RT-PCR. INHA expression was significantly elevated in 0.2% $O_2$ (4.9 times) (Fig. 1ai) in both HEY cells and OV90 cells. In OV90 cells, INHA was elevated at 1% (2.7-times) as well, however not significantly (Fig. 1ai). A similar pattern was observed for VEGFA expression with significant increases in both HEY and OV90 at 0.2% $O_2$ (HEY: 3.8 times and OV90: 4.3 times) and at 1% in HEY (2.4-times) (Fig. 1aii). HIF-1 stabilization was evaluated by western blotting to confirm an active hypoxic response that was oxygen tension dependent (Fig. 1aiii). To further test the impact of hypoxia on INHA expression, a panel of ovarian cancer cell lines representing a broad spectrum of ovarian cancer subtypes, including HEY, OV90, OVCAR5 of high grade serous origin, and PA1 a teratocarcinoma cell line of the ovary were grown for 12 or 24 h under either hypoxic conditions (0.2% $O_2$) or normoxic control tissue culture conditions (17–21%). We find a 3–6 times increase in INHA expression across all four cell lines (HEY: 4-times, OVCAR5: 4.4-times, PA1: 5.28, OV90: 4.8 times, Fig. 1bi). All cell lines showed maximum INHA increases after 24 h of hypoxia growth except for OVCAR5 which increased INHA expression within 12 h under hypoxia. VEGFA was evaluated side by side as a positive control and representative of the hypoxia response in all four cell lines and was elevated 2–6 times (HEY: 3.5 times, OVCAR5: 3.1 times, PA1: 5.18-times, OV90: 2.5 times, Fig. 1bii). A mouse ovarian cancer cell line, ID8ip2, was also tested and INHA expression was elevated 4-times here as well (Supplementary Fig. 1ai). HIF-1α stabilization in all cell lines confirmed by westerns indicated an active response to hypoxia (Fig. 1biii, Supplementary Fig. 1aii).

INHA translates into the protein inhibinα which can be secreted as a free monomer or can dimerize with INHBA or INHBB to produce dimeric functional inhibin A or inhibin B[12]. Thus, total inhibin ELISA (enzyme-linked immunosorbent assay), specific to inhibinα so as to detect all three inhibin forms, was used to test if the changes in INHA mRNA resulted in alterations to secreted protein. We find that conditioned media collected from HEY and OV90 exposed to hypoxia increased total inhibin

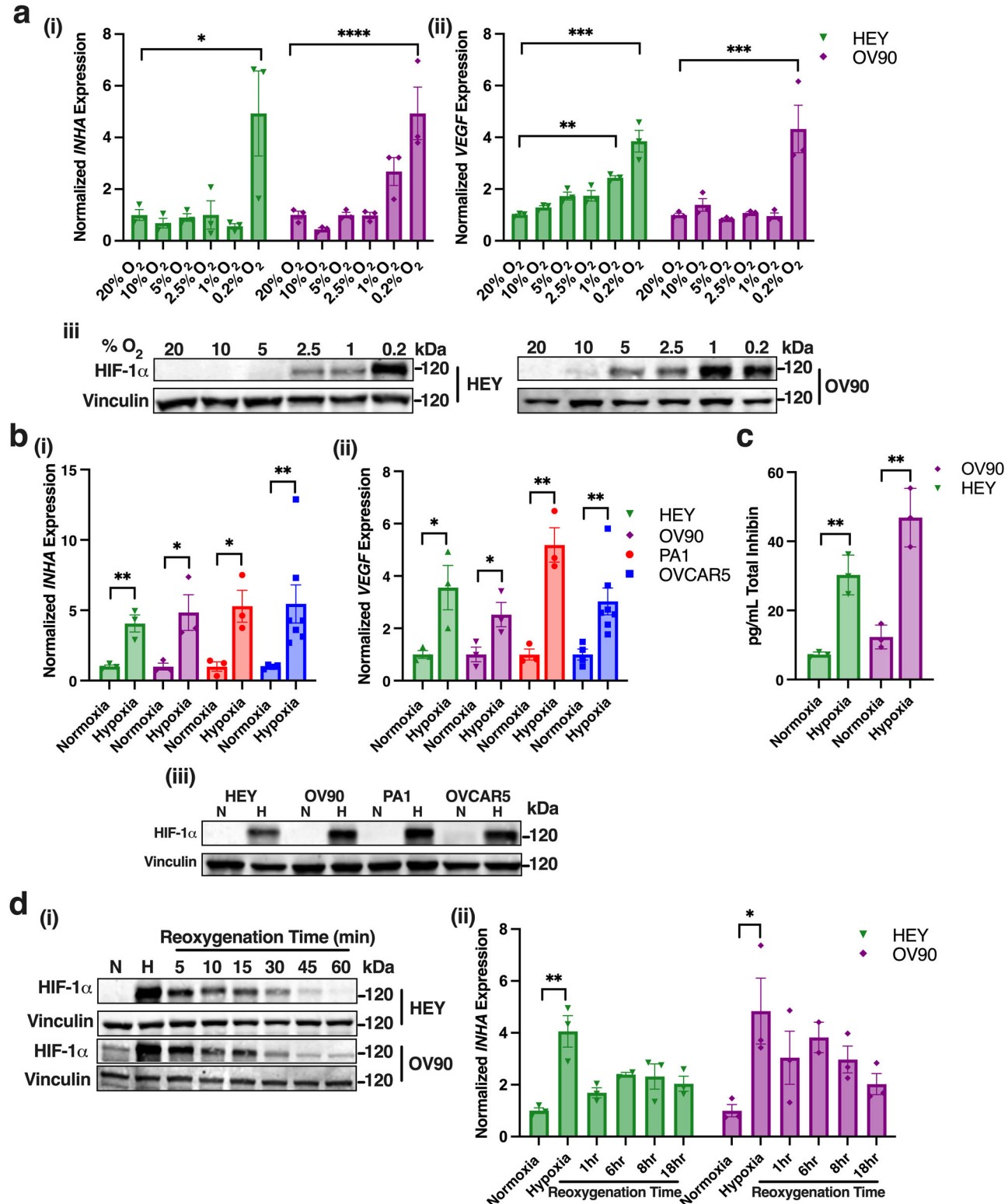

protein secretion as well, (4.2-times in HEY and 3.8 times OV90, Fig. 1c). These data suggest that *INHA* mRNA and functional secreted inhibin protein, is increased by hypoxia.

Since total inhibin protein, reflecting either inhibin A/B and free inhibinα, increased in response to hypoxia (Fig. 1c), we evaluated mRNA changes in *INHBA and INHBB* subunits in HEY and OV90 cells. While *INHA* was increased three to five-times in response to hypoxia (Fig. 1b), *INHBA and INHBB* levels were unchanged in the two cell lines evaluated (Supplementary

Fig. 1b), indicating that changes in inhibin protein levels (Fig. 1c) were largely related to increases in inhibinα. The *INHA* response to hypoxia was also more robust in tumor cells as compared to endothelial cells (HMEC-1) grown under hypoxia (0.2% O₂) for either 12 h or 24 h (Supplementary Fig. 1c) indicating that inhibinα increases in response to hypoxia occur more significantly in tumor cells.

To test if *INHA* expression remains elevated after re-exposure to oxygen, we first determined how long HIF-1 protein remained

**Fig. 1 INHA and total inhibin protein are increased in response to hypoxia in ovarian cancer cells. a** Relative qRT-PCR analysis of (i) *INHA* and (ii) *VEGFA* mRNA expression normalized to levels in 20% $O_2$ in HEY and OV90 cells exposed to indicated oxygen concentration for 24 hrs. Mean ± SEM, ($n = 3$). *$p < 0.05$; **$p < 0.01$; ***$p < 0.001$; ****$p < 0.0001$, One-way ANOVA followed by Tukey's multiple comparison. (iii) Western blot of HIF-1α stabilization at indicated oxygen concentrations in HEY (left) and OV90 (right). **b** Relative qRT-PCR analysis of (i) *INHA* and (ii) *VEGFA* mRNA expression normalized to corresponding levels in normoxia in indicated cells grown under hypoxia (0.2%) or normoxia (17–21%) for 24 h except for OVCAR5 (12 h). Mean ± SEM, n of independent trials for PA1 = 3, OVCAR5 $n = 7$, HEY $n = 3$, OV90 $n = 3$, *$p < 0.05$; **$p < 0.01$, unpaired *t*-test. (iii) Western blot of HIF-1α stabilization in indicated cell lines. **c** Total inhibin ELISA (inhibin A/B, inhibinα) of conditioned media collected from OV90 and HEY cells grown in normoxia or after 24 h exposure to hypoxia (0.2% $O_2$). Mean ± SEM, ($n = 3$). **$p < 0.01$, unpaired *t*-test. **d** (i) Western blot of HIF-1α levels in HEY and OV90 following exposure to hypoxia (0.2% $O_2$) for 24 h and after indicated reoxygenation times. (ii) Relative qRT-PCR analysis of *INHA* expression in HEY and OV90 cells following exposure to hypoxia (0.2% $O_2$) and indicated reoxygenation time normalized to corresponding levels in normoxia. Mean ± SEM, ($n = 3$). *$p < 0.05$; **$p < 0.01$, One-way ANOVA followed by Tukey's multiple comparison.

stabilized in cells when returned to normoxic conditions (reoxygenation) after 24 h exposure to hypoxia. HIF-1 protein began to decrease 5 min after re-exposure to hypoxia (reoxygenation) and went back to baseline at 60 min in HEY and OV90 cells (Fig. 1di). Since we observed HIF-1 levels return to baseline after 60 min, we began our time course for testing *INHA* expression after reoxygenation at 1 h. In HEY cells, *INHA* expression was increased four-times upon exposure to hypoxia (Fig. 1dii). Upon 1 h of reoxygenation, *INHA* expression decreased significantly in both cell lines and was no longer statistically different from normoxia grown cells (Fig. 1dii). Slight elevation in *INHA* levels remained, particularly in OV90 cells for the duration of the time course (Fig. 1dii) that did not however reach statistical significance. Taken together, these data strongly indicate that inhibinα mRNA and protein expression is increased under hypoxia conditions.

**Inhibinα is increased in ovarian cancer spheroids, patients, and tumor xenografts.** To evaluate other pathologically relevant hypoxic conditions pertinent to ovarian cancer growth and metastasis, we evaluated hypoxia and *INHA* expression in cells grown in spheroids under anchorage independence, an environment that is often hypoxic[25]. PA1 and OVCA420 cells were chosen due to their ability to form spheroids[26,27]. Cells were grown on poly-hema coated plates for either 72 h (PA1) or 48 h (OVCA420). Under such anchorage independent conditions (referred to as 3D), where HIF-1α was stabilized (Fig. 2ai), *INHA* was increased 7.8 times in PA1 and 4.6 times in OVCA420 when compared to 2D growth conditions in a dish (Fig. 2aii).

Previous studies have established that in healthy premenopausal women, inhibin levels cycle across the menstrual cycle reaching a peak of 65.6 pg/mL, while in post-menopausal women, total serum inhibin levels are below 5 pg/mL[28]. Ovarian cancer patients are commonly post-menopausal[29] and tumor tissues can display higher inhibin levels[19]. We thus wanted to assess if the peritoneal ascites fluid of advanced ovarian cancer patients, which has been shown to be a hypoxic environment[30] and contains disseminated ovarian cancer spheroids[22], also displays detectable or elevated inhibin levels. To test if inhibin protein is secreted and detectable in clinical ascites, total inhibin ELISA was performed on a cohort of 25 patient ascites. We find total inhibin levels in the range of 6.7 to 120.53 pg/mL in the ascites fluid indicating the presence of inhibin protein in ascites fluid (Fig. 2b).

We next evaluated if *INHA* expression was elevated in vivo with increasing xenograft tumor size. 5 million HEY cells were subcutaneously implanted and harvested at varying tumor sizes. Tumors greater than 500 mm³ were found to be hypoxic based on pimonidazole staining which has a detection threshold of below 10 mmHg $O_2$, or 1.2% $O_2$ (4.8 times, Fig. 2ci and Supplementary Fig. 2a)[31]. *INHA* expression was increased 9.8 times in tumors greater than 500 mm³ as compared to tumors less than 500 mm³ (Fig. 2cii). *INHA* expression was also significantly correlated with tumor size (Supplementary Fig. 2b). To further examine the

potential clinical relevance of inhibinα expression in response to hypoxia, we analyzed the TCGA/PanCancer Atlas patient data set from cBioportal[32,33] and obtained hypoxia scores from two different hypoxia gene signatures (Buffa and Winter)[34,35]. The signatures consisted of 51 (Buffa) and 99 (Winter) hypoxia related genes from a large meta-analysis of breast and head and neck squamous cell cancer that were independently verified for prognostic value[34,35]. Using these signatures, inhibinα (*INHA*) expression was significantly correlated with both hypoxia Buffa ($r = 0.1961$, $p = 0.0221$) and Winter hypoxia ($r = 0.223$, $p = 0.009$) scores in the ovarian cancer data set (Fig. 2di–ii). Analysis of breast cancer data revealed a similar trend as *INHA* expression was significantly correlated ($r = 0.2026$, $p = 0.0165$) with the Winter hypoxia score (Fig. 2diii). Taken together, these data strongly indicate that inhibinα mRNA and protein expression are increased under hypoxia conditions in ovarian cancer cell lines, xenograft tumors and in patients.

**INHA is a direct HIF-1 target under hypoxia.** Hypoxia inducible factors (HIFs) are key transcriptional regulators of the hypoxia adaptive response and increase expression of critical pro-angiogenic genes[21]. To test whether HIF proteins are regulators of *INHA* expression, we first utilized cobalt chloride (CoCl₂), a well characterized chemical stabilizer of HIF's[36]. HIF-1α was stabilized in PA1 and OVCAR5 cells treated with 100 μM of CoCl₂ for either 6, 12, or 24 h (Fig. 3ai). We find that *INHA* expression was significantly increased; 10-times in OVCAR5 after 12 h and 11.5 times in PA1 cells after 24 h of CoCl₂ treatment (Fig. 3aii). Maximum increases in *INHA* expression with CoCl₂ occurred at the same time points as exposure to hypoxia (12 h for OVCAR5 and 24 h for PA1, Fig. 1bi). *VEGFA*, used as a positive control increased 4.8 and 4.3 times at 12 h and 2.7 and 3.7-times at 24 h in both OVCAR5 and PA1, respectively (Fig. 3aii). To test if *INHA* could be a direct hypoxia target leading to increased inhibinα expression, we evaluated the effect of reducing the levels of HIF-1β/*ARNT* which is the binding partner for all HIF's[37]. Stable *ARNT* knockdown cells were generated in HEY cells (Methods). We find that control HEY cells increase *INHA* levels 2.8 times under 0.2% $O_2$ (Fig. 3b). However, shRNA *ARNT* lead to a 2.7-times reduction in hypoxia induced increase in *INHA* mRNA levels (Fig. 3b) indicating direct contributions of HIFs' to the regulation of inhibin.

To determine the roles of the HIF-1 and HIF-2 heterodimeric transcriptions factors, that both require ARNT[37], in the transcriptional regulation of *INHA* we used siRNA to knockdown the levels of HIF-1α and HIF-2α in two ovarian cancer cell lines (OV90 and HEY). Knockdown of HIF-1α and HIF-2α using siRNAs to each isoform individually or a combination of siRNAs was confirmed through western blotting (Fig. 3ci, ii). Knockdown of HIF-1α decreased hypoxia induced *INHA* expression 2.1 times in HEY and 1.9 times in OV90 compared to siScr (siHIF-1α; Fig. 3ci, ii). siRNA to HIF-2α did not result in a significant change

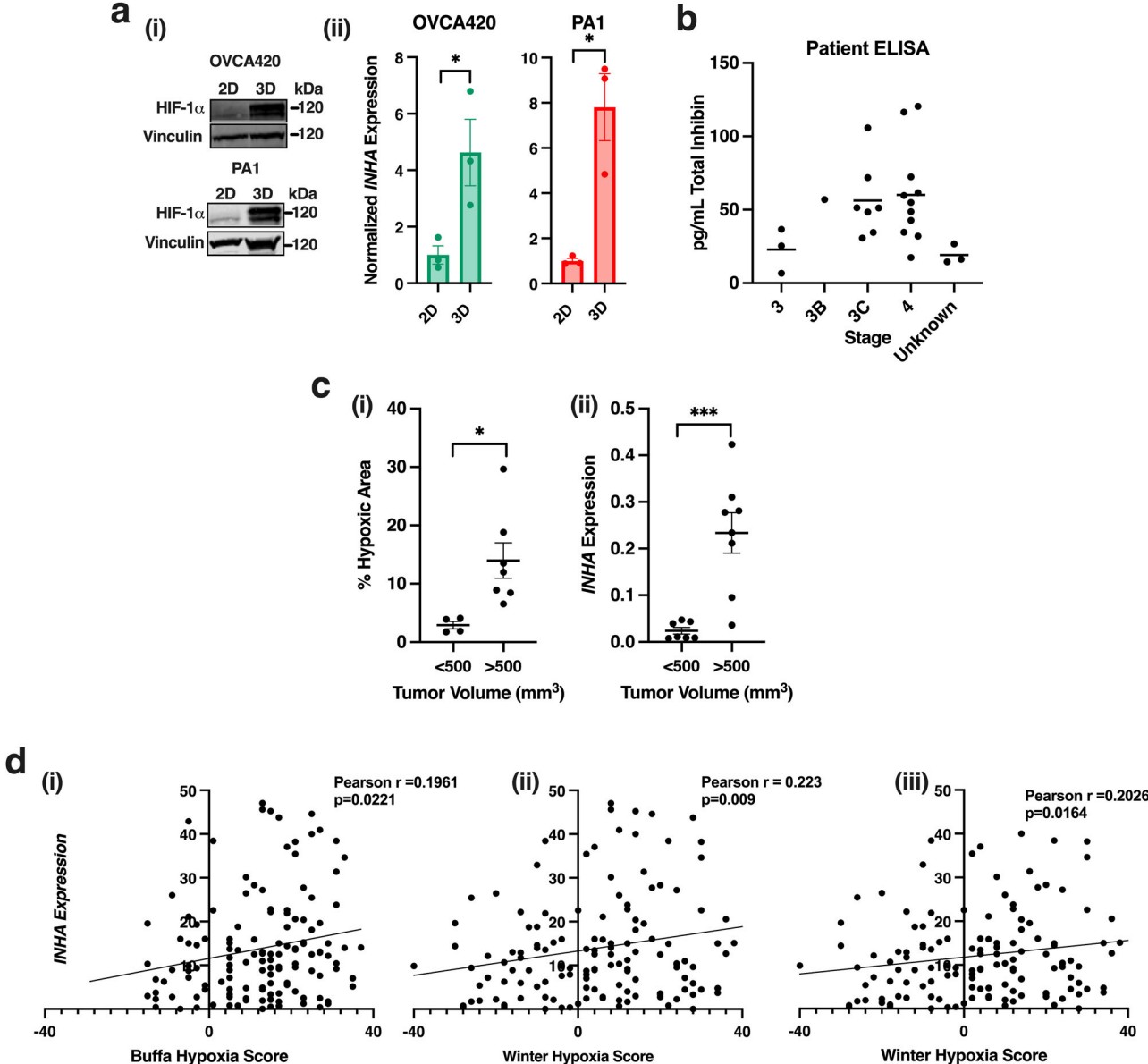

**Fig. 2 Inhibinα is increased in ovarian cancer spheroids, patient samples, and tumor xenografts. a** (i) Western blotting of HIF-1α protein from indicated cells grown in either 2D or under anchorage independence (3D) conditions, vinculin is loading control and (ii) relative qRT-PCR of *INHA* mRNA expression in OVCA420 and PA1 after 48 h (OVCA420) or 72 h (PA1) of growth under anchorage independence (3D). Mean ± SEM, $n = 3$. **$p < 0.01$; ***$p < 0.001$, unpaired *t*-test. **b** Total inhibin ELISA of ascites fluid from 25 ovarian cancer patients sorted by stage. **c** (i) Percent hypoxic area in tumors of indicated size range determined by quantitation of pimonidazole staining in tumors (Supplementary Fig. 2a). Graph represents average hypoxic area of all HEY xenograft tumors sorted by size as <500 mm³ or >500 mm³. Mean ± SEM, $n = 4$ for <500 mm³ and $n = 7$ for >500 mm³. *$p < 0.05$, unpaired *t*-test. (ii) Relative qRT-PCR of *INHA* expression in tumors from indicated sizes of HEY cells implanted subcutaneously. Mean ± SEM, $n = 8$. ***$p < 0.001$, unpaired *t*-test. **d** Correlation analysis between *INHA* expression and either (i) Buffa or (ii) Winter hypoxia scores from TCGA OVCA (i–ii) or breast (iii) cancer patient data sets from cBioportal measured by RNA-Seq. Correlation analysis was performed by Pearson correlation.

in hypoxia induced *INHA* expression compared to control (siHIF-2α; Fig. 3ci, ii). To further test that HIF-1 was the predominant HIF isoform required for hypoxia induced *INHA* expression, we utilized a double knockdown of HIF-1α and HIF-2α. In HEY and OV90 cells the double knockdown resulted in a 2-times and 1.8 times decrease in *INHA* expression compared to siScr, respectively (siHIF-1,2α; Fig. 3ci, ii). These data suggest that increases in *INHA* under hypoxia were more significantly impacted by HIF-1 as compared to HIF-2.

In silico analysis of the *INHA* gene, which is located at Chr:2q35 revealed two hypoxia response element (HRE) consensus sites within 2Kb of the promoter, GGCGTGG and

CGCGTGG, at −144 and −1789 bp from the transcription start site (TSS) (Supplementary Fig. 3a) respectively. These HRE sites conform precisely to the (G/C/T)(A/G)CGTG(G/C) consensus sequence[37]. Two hypoxia ancillary sequences (HAS) (CAGGG and CACGG) were also found directly flanking the proximal HRE sequence at −169 and −173 bp from the TSS, respectively. One HAS sequence (CACGT) was found flanking the distal HRE sequence at −1761 bp from TSS (Supplementary Fig. 3a). A previously well characterized CREB binding site (CRE) is designated for reference (Supplementary Fig. 3a).

To test direct interactions between HIF-1 and the *INHA* promoter, chromatin immunoprecipitation (ChIP) was performed

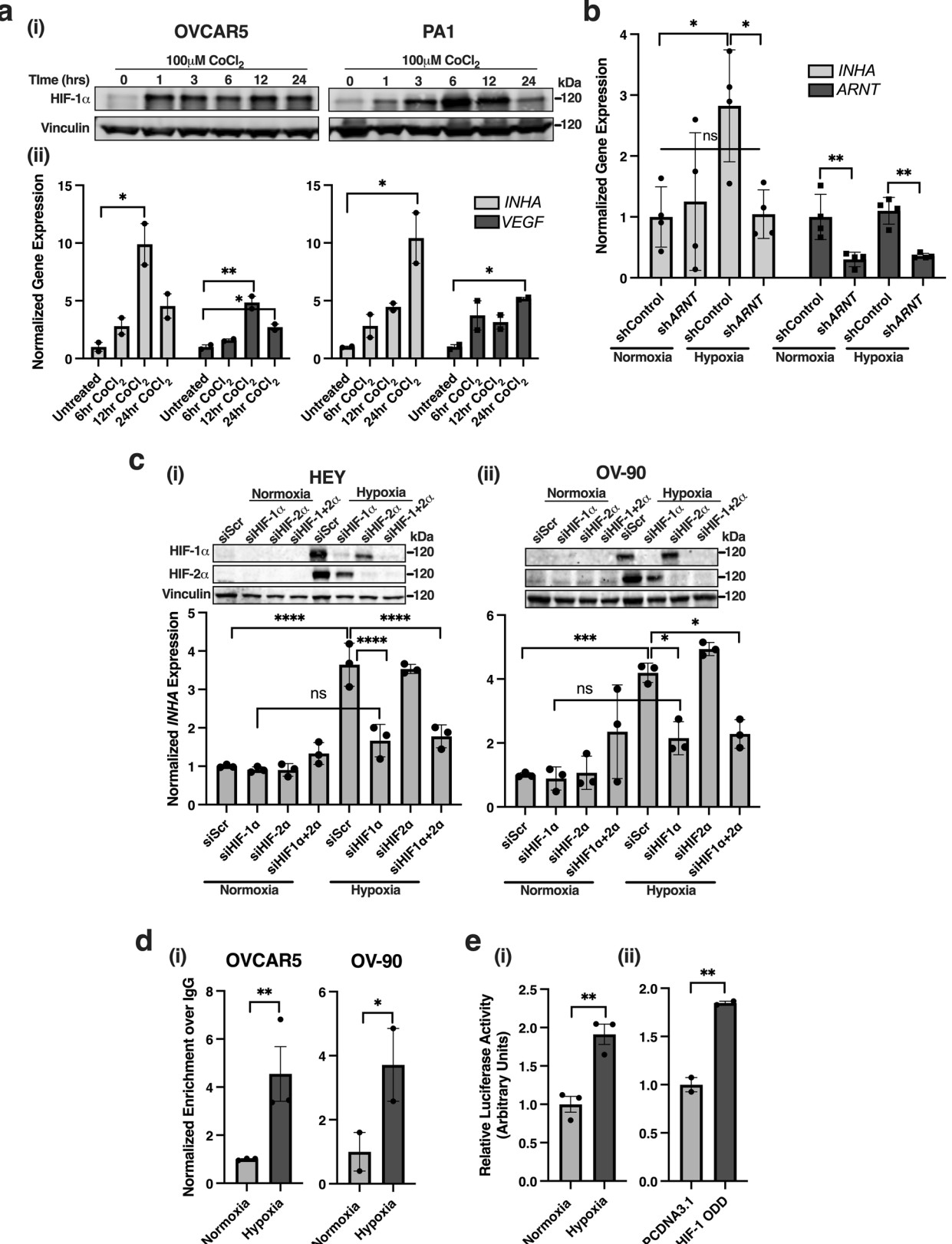

using OVCAR5 and OV90 cells. Primers were designed to amplify the region including the HRE site closest to the transcription start site (HRE1) and chromatin shear size optimized accordingly (Methods). We find that exposure to hypoxia led to a 4-times increase in enrichment of HIF-1 binding to *INHA's* HRE site in OVCAR5 and 3 times in OV90 (Fig. 3d). The second HRE site is

GC rich which led to modest amplification. Despite this, a 2-times increase in HIF-1 enrichment at this site in OV90 cells was observed (Supplementary Fig. 3b) which was however not statistically significant.

Given the poor enrichment of HIF-1 at the distal promoter site (Supplementary Fig. 3b), we next evaluated if the proximal

**Fig. 3 *INHA* expression is regulated by HIF-1. a** (i) Western blot of HIF-1α at indicated time points after treatment with 100 μM CoCl₂. (ii) Relative qRT-PCR analysis of *INHA* and *VEGF* mRNA in OVCAR5 and PA1 cells after indicated time of treatment with 100 μM of CoCl₂ normalized to untreated. Mean ± SEM, (*n* = 2). *p < 0.05; **p < 0.01, One-way ANOVA followed by Tukey's multiple comparison. **b** Relative qRT-PCR analysis of *INHA* and *ARNT* mRNA in HEY shControl or shARNT cell lines after exposure to hypoxia (0.2% O₂) for 24 h normalized to corresponding shControl normoxia levels. Mean ± SEM, (*n* = 4). n.s.,not significant; *p < 0.05; **p < 0.01, unpaired *t*-test. **c** Representative western blot (above) and relative qRT-PCR analysis of *INHA* expression (below) from (i) HEY or (ii) OV90 cells transfected with either siScr, siHIF-1α, siHIF-2α, or a combination of siHIF-1/2α and exposed to hypoxia (0.2% O₂) for 24 h. Mean ± SEM, (*n* = 3) *p < 0.05; ***p < 0.001; ****p < 0.0001, One-way ANOVA followed by Tukey's multiple comparison test. **d** Relative qRT-PCR analysis using primers that amplify the proximal HRE region in the *INHA* promoter (Supplementary Fig. 3a) after chromatin immunoprecipitation (ChIP) of HIF-1α in OVCAR5 and OV90 cells. ChIP qRT-PCR results were quantified as normalized enrichment over IgG and normalized to normoxia. Mean ± SEM, OVCAR5 (*n* = 3), OV90 (*n* = 2). n.s.,not significant; *p < 0.05; **p < 0.01, Two-way ANOVA followed by Fishers LSD test. **e** Luciferase activity of HEK293 cells transfected with the *INHA* promoter driven luciferase reporter construct (pGL4.10) and a SV-40 renilla control vector. Cells were either (i) exposed to hypoxia (0.2% O₂) or (ii) co-transfected with HIF-1α overexpression plasmid (HIF-1 ODD) and luciferase activity measured and normalized to either normoxia in (i) or PCDNA3.1 in (ii). Mean ± SEM, *n* = 3 (Hypoxia), *n* = 2 (HIF-1 ODD) *p < 0.05; **p < 0.01, unpaired *t*-test.

promoter was sufficient to increase *INHA* levels under hypoxia and if this was dependent on HIF-1. To achieve this, we made a *INHA* promoter driven luciferase reporter construct, containing 547 base pairs of the *INHA* promoter, including the first HRE site (Fig. 3e, Supplementary Fig. 3a). The effect of HIF-1 on *INHA* promoter activity, was evaluated in HEK293 cells exposed to hypoxia (0.2% O₂) for 24 h and compared to cells under normoxia (Fig. 3ei), or in the presence or absence of HIF-1 ODD (pcDNA3-HA-HIF1aP402A/P564A) (Fig. 3eii) that prevents degradation of the HIF-1α subunit[38]. We find that in untransfected or control vector expressing cells (pcDNA3.1), *INHA* promoter driven luciferase activity is increased two times in response to hypoxia (Fig. 3ei) that was mimicked by stabilization of HIF-1α (HIF-1 ODD) under normoxia conditions (Fig. 3eii). We also confirmed the requirement of HIF-1α in HEK293 using either control or HIF-1/2α siRNAs. HEK293 were exposed to hypoxia for 24 h and efficacy of HIF-1/2α knockdown was confirmed by immunoblotting (Supplementary Fig. 3ci). Notably, siRNA to HIF-1α (siHIF-1α) decreased hypoxia induced *INHA* expression 1.8 times as compared to scramble controls (siScr; Supplementary Fig. 3cii). However, siRNA to HIF-2α resulted in a smaller (1.25-times) and non-significant reduction in *INHA* expression compared to siScr when exposed to hypoxia (Supplementary Fig. 3cii). These data point to a central role for HIF-1 in regulating *INHA* expression under hypoxia.

*INHA* has been previously reported to be regulated by other factors particularly the cAMP response element binding (CREB) family member in multiple systems[8]. The CREB family of transcription factors can act downstream of the hypoxia response[39]. To thus test whether cAMP was involved in regulating *INHA* expression under hypoxia, we utilized forskolin (Fsk), an activator of cAMP previously shown to induce *INHA* expression and the PKA inhibitor, H89, previously shown to inhibit forskolin induced *INHA* expression[40]. Treatment of ID8ip2 cells with Fsk increased *INHA* expression 5.2-times under hypoxia compared to just 2-times under normoxia (Supplementary Fig. 3di). This relationship appeared to be additive and not synergistic as addition of the PKA inhibitor, H89, was not able to reduce hypoxia induced *INHA* expression (Supplementary Fig. 3di). The effect of blocking PKA signaling under hypoxia was also tested in the ovarian cancer cell line OV90. While hypoxia increased *INHA* expression 4.5 times in OV90 cells, treatment with H89 did not significantly reduce *INHA* expression under hypoxia (Supplementary Fig. 3dii). Taken together, these data implicate HIF-1 as being the key transcriptional factor responsible for increase of *INHA* in hypoxia.

**Inhibin promotes hypoxia induced angiogenesis and stimulates endothelial cell migration and vascular permeability.** Hypoxia is a key driver of endothelial cell migration and blood vessel permeability within the tumor leading to alterations in angiogenesis[24]. To determine the overall contribution of inhibin to hypoxia induced angiogenesis in vivo, we utilized an in vivo Matrigel plug assay. Conditioned media (CM) from HEY tumor cells exposed to normoxia or hypoxia was used to stimulate angiogenesis into the plugs, and a well-established anti-inhibinα antibody, R1 (recognizing the junction between the αN region, and αC region)[41] was used to block inhibin in the CM with IgG as a control. We find that CM from hypoxia grown cells increased hemoglobin in the plugs 2.9 times compared to CM from normoxia grown cells (Fig. 4ai–ii). Anti-inhibinα in the hypoxic CM fully reduced the hemoglobin content in the plug (2.1 times suppression, Fig. 4ai–ii) indicating that inhibin is required for hypoxia induced blood vessel formation in vivo.

Since blood vessel flow is an indication of endothelial cell functionality[42], we sought to define the specific effects of increased inhibinα on hypoxia induced endothelial cell biology, specifically endothelial cell chemotaxis and vascular permeability. To determine the impact on endothelial chemotaxis to hypoxic CM, CM from either hypoxia (24 h, 0.2% O₂) or normoxia grown OV90 or HEY cells were used as a chemoattractant to measure migration of human microvascular endothelial cells (HMEC-1; Fig. 4b). Two anti-inhibinα antibodies, R1 and a second well-established antibody PO23 (recognizing the C-terminus of the αC region)[41], were used with IgG controls to test the effect of blocking/sequestering hypoxia produced inhibinα. We find that CM from hypoxia grown tumor cells significantly increased migration of endothelial cells (IgG, Fig. 4b) and incubation of hypoxic CM with anti-inhibinα R1 significantly suppressed hypoxia induced endothelial migration (2.1 and 1.6 times for OV90 and HEY conditioned media respectively, Fig. 4bi–ii). Anti-inhibinα PO23 was also able to significantly suppress CM stimulated endothelial migration (1.5 and 1.75-times for OV90 and HEY CM, respectively, Fig. 4bi–ii). Similar to the effects of hypoxic CM, recombinant inhibin A was also able to stimulate HMEC-1 migration to similar extents as *VEGFA* at equimolar amounts (Fig. 4biii).

We next evaluated the effect of CM from hypoxic tumor cells on changes to permeability across an endothelial monolayer using a trans-well permeability assay that measures solute (FITC-dextran) flux across endothelial monolayers. Permeability was monitored across a 4-h time course and CM from hypoxic tumor cells was used to induce permeability across the HMEC-1 monolayer. Effect of inhibin in the CM was evaluated either in the presence of anti-inhibinα (PO23 and R1) or IgG control (Fig. 4ci–ii). We find that both inhibinα antibodies (R1 and PO23) significantly decreased solute flux induced by hypoxic CM from two tumor cell lines, albeit with moderate differences in the kinetics and time to inhibition (Fig. 4ci–ii). Specifically, significant inhibition of permeability was seen beginning at 2 h

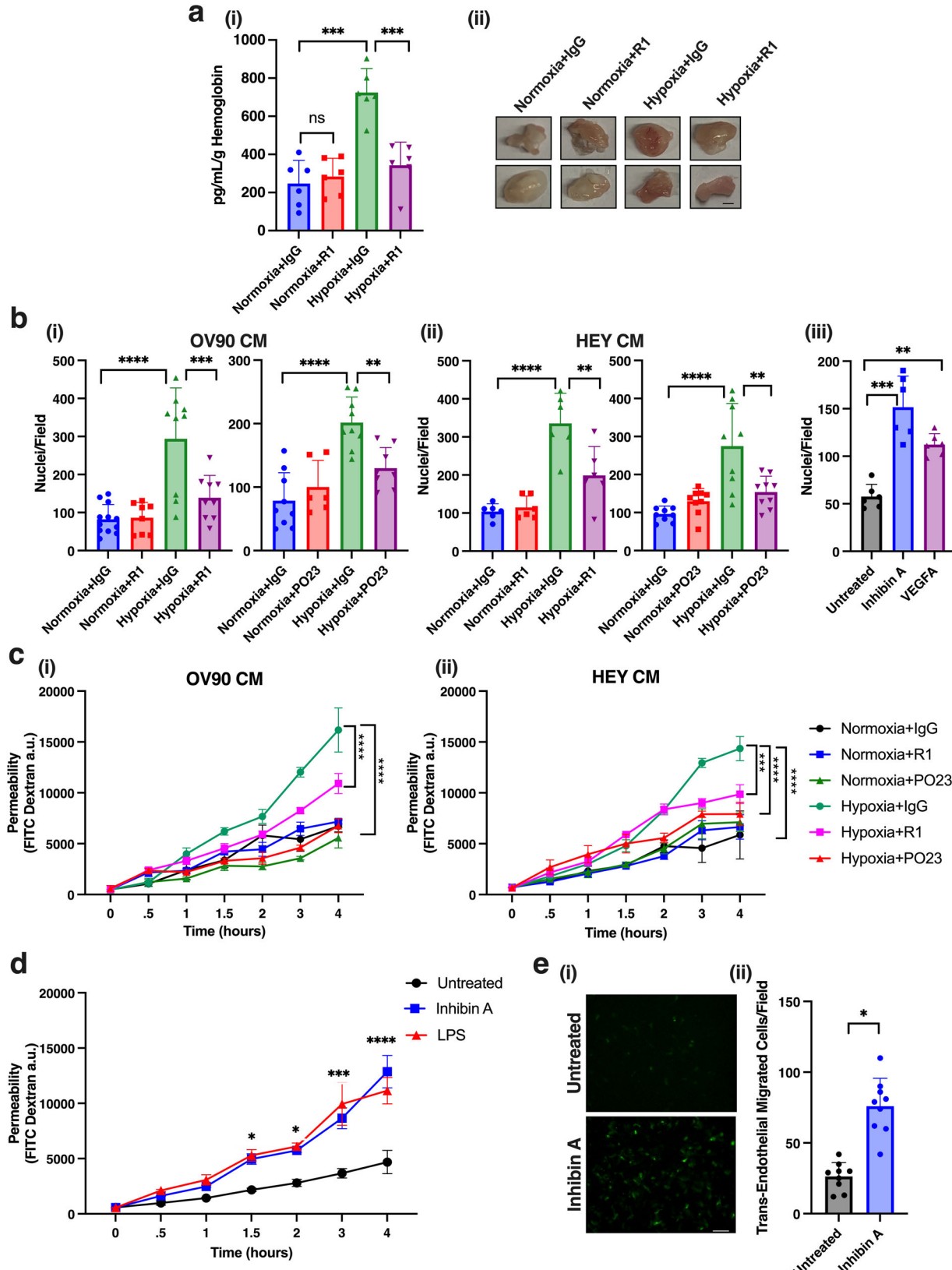

for CM treated with PO23 and 3 h for R1. PO23 was moderately more effective than R1 as it effectively reduced permeability within 1 h (Fig. 4ci–ii). Recombinant inhibin was also able to induce endothelial cell permeability to similar extents as lipopolysaccharide (LPS) (Fig. 4d), an established permeability inducing factor[30]. Since perturbations to the endothelial barrier

are critical to invasion and extravasation of cancer cells during metastasis[43], we tested whether inhibin induced vascular permeability facilitates tumor cell extravasation. To test this, we used a trans-endothelial cell migration assay to mimic the process. HEY tumor cells infected with GFP adenovirus to distinguish them from migrated non-GFP endothelial cells were plated on top of a

**Fig. 4 Inhibin increases hypoxia induced angiogenesis and endothelial cell migration and permeability in vivo and in vitro respectively. a** (i) Hemoglobin content in Matrigel plugs collected 12 days after subcutaneous injection of HEY conditioned media collected from cells exposed to normoxia or hypoxia for 24 h and mixed with either 2 μg of IgG or anti-inhibin R1 antibody. Mean ± SEM, $n = 6$ plugs per condition. n.s., not significant; ***$p < 0.001$, One-way ANOVA followed by Tukey's multiple comparison test. (ii) Representative images of Matrigel plugs from (i) Scale bar: 2 mm. **b** Quantitation of HMEC-1 migration through fibronectin coated 8 μm trans-well filter (i–ii) towards conditioned media from OV90 or HEY cells exposed to hypoxia (0.2% $O_2$) with either 2 μg of R1 or PO23 anti-inhibin antibody or IgG as a control, or towards (iii) serum free media containing 1 nM inhibin A or 1 nM VEGFA. Nuclei from three representative fields per filter were counted. Mean ± SD. **$p < 0.01$; ***$p < 0.001$; ****$p < 0.0001$, One-way ANOVA followed by Tukey's multiple comparison. **c**, **d** Quantitation of endothelial cell permeability by measuring FITC-dextran changes across a HMEC-1 monolayer treated with (i–ii) conditioned media from (i) OV90 or (ii) HEY cells exposed to hypoxia (0.2% $O_2$) with either 2 μg of R1 or PO23 anti-inhibin antibody or IgG as a control, or (**d**) treated with 1 nM inhibin A or 10 μg/mL LPS. Mean ± SEM *$p < 0.05$; ***$p < 0.001$; ****$p < 0.0001$, One-way ANOVA followed by Tukey's multiple comparison. **e** HEY trans-endothelial migration (TEM) across HMEC-1 monolayer either treated with inhibin A for 4 h or untreated. (i) Representative transmigrated GFP positive HEY cells and (ii) quantitation of transmigration ($n = 3$). *$p < 0.05$, unpaired $t$-test. Scale bar: 100 μm.

non-GFP endothelial cell monolayer that was then either pre-treated with 1 nM inhibin A for 4 h or left untreated. We find that HEY GFP tumor cells were 2.9 times more invasive across the inhibin treated monolayer than untreated conditions (Fig. 4ei–ii). All together, these data implicate inhibin as a robust contributor to hypoxia mediated angiogenesis, vascular permeability and thereby tumor cell extravasation across the vascular endothelium.

**Inhibin promotes vascular permeability through increased VE-cadherin trafficking.** Endothelial permeability is regulated through changes in junctional proteins which are maintained through contacts with the actin cytoskeleton[44]. VE-cadherin is a critical junctional protein involved in regulating endothelial cell permeability[44]. To delineate the mechanism of inhibin's effects on vascular permeability, we first evaluated the effect of inhibin on endothelial cell junctions and the actin cytoskeleton through immunofluorescent staining of VE-cadherin and actin (Fig. 5a). Examination of the actin cytoskeleton revealed contractile actin staining, with a quantifiably significant increase in stress fiber formation after 30 min of inhibin A treatment (two times increase, Fig. 5ai–ii). VEGFA treatment was used as a comparison that also led to similar changes in actin stress fiber formation (Fig. 5a). VE-cadherin localization also appeared to be reduced qualitatively at the cell-cell junctions after 30 min of inhibin treatment as compared to untreated cells, suggestive of perturbation of the endothelial cell barrier at the level of the cytoskeleton (Fig. 5a). Loss of VE-cadherin at the cell junctions was also observed in VEGFA treated cells (Fig. 5a). However, total VE-cadherin levels were unchanged in response to inhibin as evaluated over a time course of 60 min (Supplementary Fig. 4a) indicating no change in the total pool of VE-cadherin in response to inhibin A. Actin contractility and stress fiber assembly is regulated through phosphorylation of myosin light chain (MLC)[44]. In accordance, we find that phosphorylation of MLC-2 (Ser19) increased within 5 min of inhibin A treatment and was sustained across a 60-min time course (Fig. 5bi–ii). Based on the qualitative changes in VE-cadherin in response to inhibin A treatment (Fig. 5a), we tested whether alterations in VE-cadherin at the cell-cell junctions were due to inhibin induced VE-cadherin internalization (Fig. 5c). To determine this, HMEC-1 membrane localized VE-cadherin was labeled at 4 °C with an anti-VE-cadherin antibody recognizing the extracellular domain. HMEC-1 cells were washed with acid to remove membrane bound anti-VE-cadherin leaving only any internalized VE-cadherin that may have been labeled at 4 °C prior to treatment with inhibin A or VEGFA (Fig. 5ci). Stripping of cell surface VE-cadherin was verified by cell surface immunostaining of VE-cadherin with little to no internalized VE-cadherin detected (Fig. 5cii, iv). Cells were then either left untreated or treated for 30 min with inhibin A at 37 °C and VE-cadherin evaluated by immunofluorescence (Fig. 5ciii). We find that inhibin A increased the internalized VE-

cadherin pool compared to untreated cells 1.4-times (Fig. 5cv) and to similar extents as VEGFA (1.6 times, Fig. 5cv). The untreated HMEC-1 had 5% of cells with internalized VE-cadherin while inhibin A and VEGFA treated HMEC-1 resulted in 54% and 42%, of cells respectively, with detectable internalized VE-cadherin (Supplementary Fig. 4b). These results indicate that inhibin induces rapid changes in the actin cytoskeleton and trafficking of VE-cadherin from the cell junctions of endothelial cells.

**Inhibin's effects on vascular permeability are mediated by ALK1 and CD105/endoglin that form a stable complex at the cell surface in response to inhibin.** Previously, we demonstrated that inhibin's effects on angiogenesis and endothelial cell signaling were dependent on the TGFβ receptors ALK1 and endoglin[19]. To evaluate if ALK1 and endoglin are required for inhibin's influence on vascular permeability, we treated HMEC-1 cells with Tracon 105 (TRC105), a humanized endoglin monoclonal antibody[45], or with ALK1-Fc, a human chimeric ALK1 protein[46] (Fig. 6a). At 4 h, treatment with (i) TRC105 and (ii) ALK1-Fc decreased inhibin A induced permeability by 2.2 and 1.5 times, respectively (Fig. 6a, Supplementary Fig. 5 for full time course) indicating both ALK1 and endoglin are required for inhibin's effects on endothelial cell permeability.

We next evaluated if internalization of VE-cadherin by inhibin was dependent on endoglin using mouse embryonic endothelial cells (MEEC) that are either wild type (WT) or null for endoglin expression[47] (Supplementary Fig. 6). Cell surface biotinylation of VE-cadherin was used to quantitatively assess VE-cadherin internalization. Towards this, cell surface proteins were labeled with Sulfo-NH-SS biotin and allowed to internalize for 30 min at 37 °C in the presence or absence of inhibin followed by stripping of cell surface biotin, immunoprecipitation with neutravidin resin and immunoblotting to detect internalized biotin labeled VE-cadherin (Fig. 6bi). Treatment with inhibin A increased internalized VE-cadherin 1.9 times in MEEC WT compared to control (Fig. 6bii), similar to extents seen by immunofluorescence in HMEC-1 cells (Fig. 5c). However, in the absence of endoglin in MEEC ENG−/− cells, inhibin A did not change the internalized VE-cadherin pool (Fig. 6bii). This data indicates that endoglin is essential for inhibin's effects on VE-cadherin.

Based on the significant dependency of inhibin's effects on endothelial cell permeability and VE-cadherin internalization on endoglin and ALK1 respectively (Fig. 6a, b), we evaluated biophysically, in a sensitive and quantitative manner, the extent of the endoglin-ALK1 interaction in response to inhibin. We utilized a patch/FRAP (fluorescence recovery after photobleaching) methodology to measure interactions between endoglin and ALK1 at the surface of live cells. This method differentiates between stable and transient interactions as described in detail previously[48]. Herein, one receptor carrying an extracellular

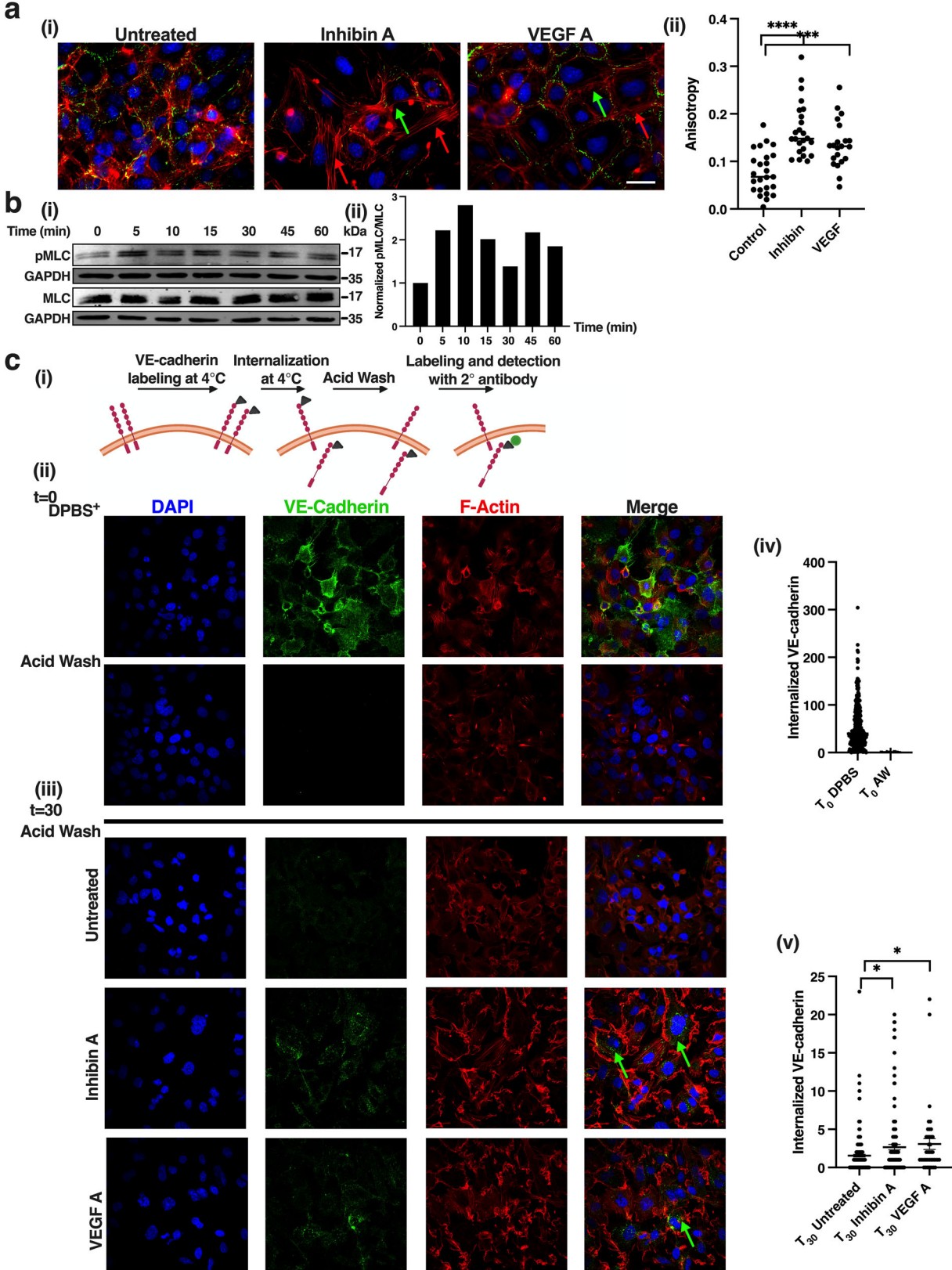

epitope tag is patched and immobilized through cross-linking with a double layer of IgGs. The effects of this immobilization on the lateral diffusion of a co-expressed, differently tagged receptor labeled exclusively with Fab' fragments are then measured by FRAP (Methods). Stable complex formation between the two co-expressed receptors (complex lifetimes longer than the characteristic FRAP fluorescence recovery time) reduces the mobile fraction ($R_f$) of the Fab'-labeled receptor, since bleached Fab'-labeled receptors associated with immobilized receptors do not appreciably dissociate from the immobile patches during the FRAP measurement. On the other hand, transient complexes (short complex lifetimes) would reduce the apparent lateral

**Fig. 5 Inhibin increases endothelial cell contractility and VE-cadherin internalization. a** (i) Representative immunofluorescence images of F-actin (red) or VE-Cadherin (green) from HMEC-1 cells grown to confluence on fibronectin coated coverslips and treated with either 1 nM inhibin A or 1 nM *VEGFA* for 30 min. (ii) Quantitation of actin stress fibers from (i) using ImageJ FibrilTool plugin. ***$p < 0.001$; ****$p < 0.0001$, unpaired *t*-test. Scale bar: 25 µm. **b** (i) Western blot analysis of pMLC-2 from HMEC-1 cells upon 1 nM inhibin A treatment for indicated times. (ii) Quantitation of pMLC-2 changes in (i). **c** (i) Schematic of VE-cadherin internalization (ii) Representative immunofluorescent images of (upper panel) cell surface labeled VE-cadherin at 4 °C detected by labeling with an extracellular domain anti-VE-cadherin antibody. Efficiency of stripping of extracellular labeled VE-cadherin with a mild acid in lower panel. (iii) Internalized VE-Cadherin at 37 °C detected with a FITC-secondary antibody in either untreated or cells treated with 1 nM inhibin A or 1 nM *VEGFA* after acid wash. Green arrows represent internalized VE-cadherin. Red, actin. Blue, DAPI. Quantitation of internalized VE-Cadherin at (iv) $T_0$ or (v) $T_{30}$ by Blobfinder ImageJ Plugin (Methods). *$p < 0.05$, unpaired *t*-test. Scale bar: 25 µm.

diffusion coefficient (*D*), since each Fab'-labeled receptor molecule can undergo multiple association-dissociation cycles during the FRAP measurement[48]. For these studies, COS7 cells were transfected with myc-ALK1, HA-endoglin or co-transfected with both, and subjected to patch/FRAP experiments in the absence or presence of 4 nM of inhibin A (Fig. 6c). Figure 6ci–iii depict representative FRAP curves showing the lateral diffusion of myc-ALK1 (Fig. 6ci), IgG-crosslinked and immobilized HA-endoglin (Fig. 6cii), and myc-ALK1 co-transfected with HA-endoglin followed by IgG cross-linking of HA-endoglin in the presence of inhibin (Fig. 6ciii). Average values derived from multiple independent experiments are shown in ($R_f$ in Fig. 6civ, *D* values in Fig. 6cv). Singly expressed myc-ALK1 had lateral mobility resembling other TGF-β superfamily receptors[49], which was insensitive to inhibin treatment (Fig. 6ci and iv). Immobilization of HA-endoglin (Fig. 6cii and iv) reduced $R_f$ of myc-ALK1 by about 45%, and the presence of inhibin increased this reduction significantly (from 45% to 70% reduction) (Fig. 6ciii and iv). Under all these conditions, the lateral diffusion coefficient (*D*) of myc-ALK1 was not significantly affected (Fig. 6cv), indicating that endoglin and ALK1 form stable complexes at the plasma membrane which are enhanced and stabilized by inhibin.

Previous studies indicate that inhibinα may bind to ALK4[50], an established Type I receptor for the activin family of proteins[8]. We thus employed patch/FRAP to determine the interactions between endoglin and ALK4 and to examine whether inhibin A enhanced these interactions. To this end, we expressed HA-ALK4, myc-endoglin or both in COS7 cells, and subjected them to patch/FRAP studies on the lateral diffusion of HA-ALK4 without and with IgG cross-linking of myc-endoglin, and with or without inhibin A. In the absence of inhibin A, endoglin and ALK4 exhibited significant stable interactions, as demonstrated by the reduction in $R_f$ of HA-ALK4 upon immobilization of myc-endoglin (40% reduction in $R_f$, with no effect on the *D* value) (Fig. 6di–ii). However, in contrast to the observations with endoglin-ALK1 complexes, the interactions between endoglin and ALK4 were weakened in the presence of inhibin A (the reduction in $R_f$ decreased to 20%) (Fig. 6di). Taken together, these results indicate that inhibin shifts the balance of endoglin complexes from interactions with ALK4 to interactions with ALK1, both of which (endoglin and ALK1) are required for inhibin-mediated vascular permeability.

**Inhibin promotes hypoxia induced tumor growth in vivo through alterations in permeability and angiogenesis.** The significance of hypoxia in ovarian cancer is well documented and we previously demonstrated increased ascites accumulation in tumor bearing mice in the presence of inhibin[19]. To precisely define the contribution of inhibin to hypoxia induced tumor growth and angiogenesis, we first evaluated the effects of pre-exposure to hypoxia on tumor growth in a subcutaneous model in vivo, a model that allows for quantitative analysis of the vasculature in tumors[51]. HEY pLKO.1 control vector (shControl) cells were pre-exposed to hypoxia (0.2% $O_2$) for 24 h or kept under normoxia followed by injection into the right flank of Ncr

nude mice. Tumors were measured throughout and harvested after 30 days ($n = 10$ mice). HEY cells pre-exposed to hypoxia produced rapid growing tumors compared to those that originated from normoxia grown cells (Fig. 7a, purple versus gray). In parallel, we utilized two methods to perturb inhibin: (1) shRNA knockdown of *INHA* in HEY cells (Supplementary Fig. 7a, b) and (2) intraperitoneal administration of anti-inhibinα antibody (R1). R1 is a human antibody[41] and consistent with this, no overall toxicity was noted in pilot toxicity studies that utilized daily injections of R1 (Supplementary Fig. 7c). sh*INHA* cells exposed to hypoxia maintained their knockdown to *INHA* at the end of the study (Supplementary Fig. 7b) and produced tumors with significantly slower growth rates than shControl hypoxia tumors (Fig. 7a, blue versus gray). In complementary findings, hypoxia exposed tumor cells had significantly reduced tumor growth upon receiving treatment with the R1 antibody when compared to tumors in mice that received vehicle only (Fig. 7a, red versus gray, $n = 6$ for R1 treated mice). The group receiving anti-inhibinα (R1) grew at a similar rate as the sh*INHA* hypoxia tumors (Fig. 7a, red versus blue). In mice with sh*INHA* tumors, treatment with R1 further reduced tumor growth albeit moderately compared to vehicle sh*INHA* (Fig. 7a, blue versus green). These data indicate that perturbation of inhibin through shRNA targeting and anti-inhibin antibody treatment reduces tumor growth. We hence sought to determine the effect of sh*INHA* on the angiogenic cytokine profile of the tumors using a proteome array of 55 different human angiogenesis targets. We find that the most upregulated proteins in control tumors compared to sh*INHA* tumors were a subset of pro-angiogenic cytokines: IL-8 (2.5 times) and EGF (2.1 times) (Fig. 7bi) indicating a pro-angiogenic profile of the tumor cells in the presence of inhibin. In contrast, the sh*INHA* hypoxia tumors showed increases in proteins including ADAMTS-1 (1.6 times) and Pentraxin-3 (1.3 times), indicating an anti-angiogenic profile in *shINHA* tumor cells as both have been demonstrated to be anti-angiogenic[52,53]. Activin A and endoglin were also found to be elevated in sh*INHA* tumors (Fig. 7bi). To complement the human tumor array, we analyzed changes in the mouse angiogenic proteome as well to delineate any host differences in response to shControl and sh*INHA* tumor cells. We find that host cells also upregulated significantly more pro-angiogenic proteins, including CXCL16[54], PIGF-2[55], and NOV[56] in shControl tumors compared to sh*INHA* tumors (Fig. 7bii). Taken together, these data suggest that altering inhibin in the tumors results in a change in the balance of angiogenic factors leading to a significant reduction in pro-angiogenic factors and slower overall tumor growth.

To rule out whether the reduction in tumor growth in sh*INHA* cells was due to slower proliferation of tumor cells, growth rate of HEY sh*INHA* and HEY shControl was evaluated in culture under hypoxia for 3 days. No significant change was observed (Supplementary Fig. 7d) suggesting that the major effect of inhibin on tumor growth are likely through effects on the tumor vasculature due to the effects of hypoxia regulated inhibin on angiogenesis and vascular permeability in vitro (Figs. 4 and 5).

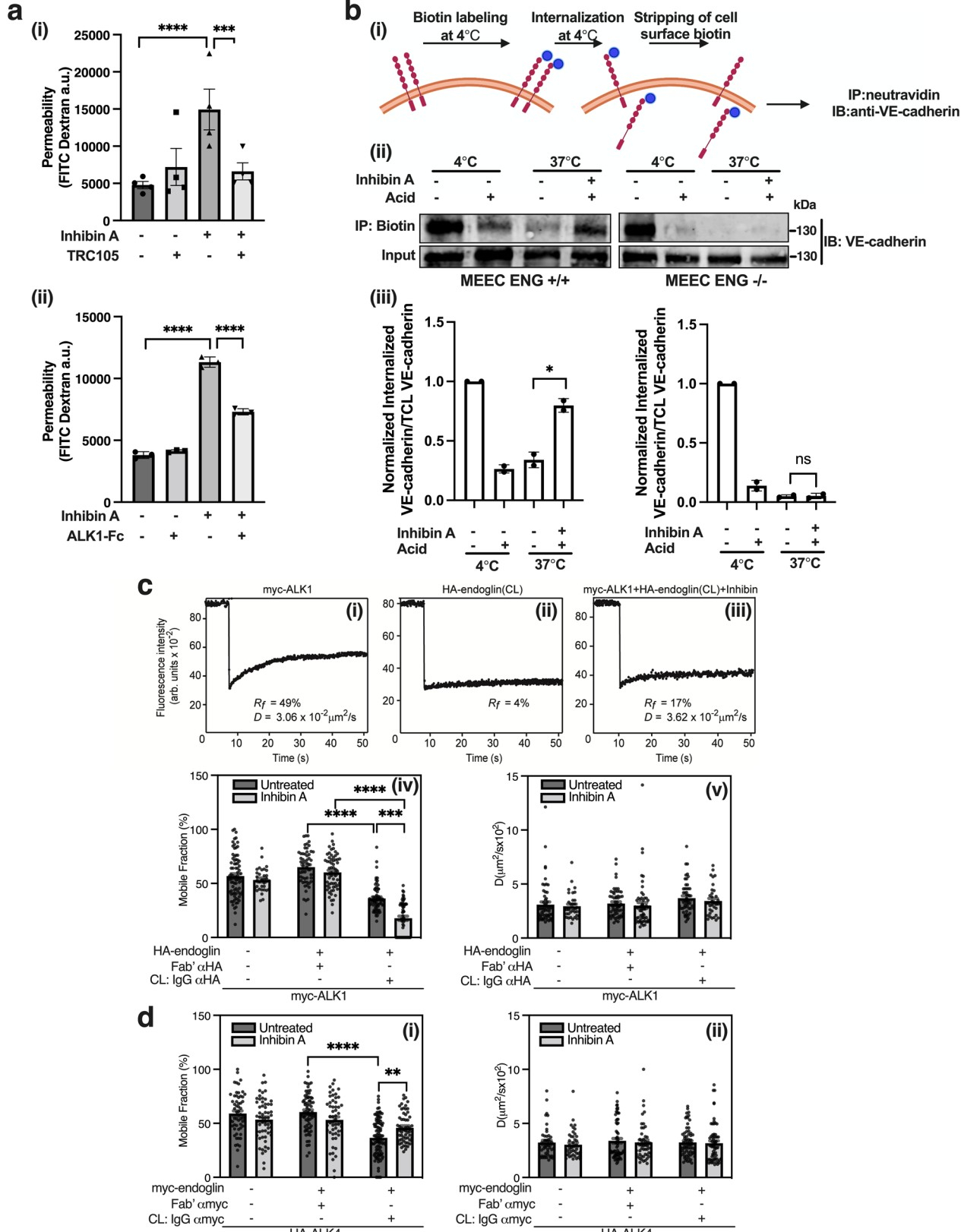

We thus determined the effect of inhibin on the tumor vasculature and associated permeability changes as a contributing factor to the altered tumor growth in sh*INHA* and antibody treated hypoxia tumors (Fig. 7a). To this end, HEY shControl or sh*INHA* cells pre-exposed to hypoxia for 24 h were injected subcutaneously into the right flank of Ncr nude mice (*n* = 4 mice)

and tumors in all groups were harvested upon reaching 700–850 mm$^3$ (Fig. 7ci) to eliminate any tumor size effects on angiogenesis. These tumors (Fig. 7ci) were evaluated for changes in vascular permeability by visualization of a rhodamine-dextran dye that leaks from the blood vessels into the tumors when administered into mice prior to sacrifice. We find that

**Fig. 6 Inhibin promotes endothelial cell permeability via ALK1 and endoglin and specifically increases ALK1-endoglin cell surface complexes while reducing ALK4-endoglin complexes. a** Quantitation of endothelial cell permeability by measuring FITC-dextran changes across a HMEC-1 monolayer treated with 1 nM inhibin A in the presence or absence of (i) 100 μg/mL TRC105 or (ii) 10 ng/mL ALK1-Fc. FITC-dextran diffusion across the HMEC-1 monolayer at 4 h is presented. Mean ± SD, $n = 4$ for i and $n = 3$ for ii. n.s. not significant; ***$p < 0.001$; ****$p < 0.0001$. **b** Internalization of VE-cadherin measured by cell surface biotinylation. (i) Biotin labeling of cell surface proteins was performed on MEEC WT or MEEC ENG−/−. Internalization was induced by treatment with 1 nm Inhibin A for 30 min at 37 °C followed by stripping of cell surface biotin. Internalized VE-cadherin was detected by IP with neutravidin resin and (ii) immunoblotting with anti-VE-cadherin and (iii) quantitated as internalized VE-cadherin over input VE-cadherin normalized to 37 °C control. Mean ± SD, $n = 2$. n.s. not significant; *$p < 0.05$, unpaired t-test. **c, d** Patch/FRAP studies on the effect of inhibin A on endoglin-ALK1 (**c**) and endoglin-ALK4 (**d**) complex formation. COS7 cells were transfected with myc-ALK1 and HA-endoglin (**c**) or with (each vector HA-ALK4 and myc-endoglin) (D) (each vector alone, or together). **c** After 24 h, singly transfected cells were labeled for FRAP by anti-tag Fab′ followed by fluorescent secondary Fab′ (Methods) and subjected to FRAP studies. For patch/FRAP, cells were subjected to protocol 1 of IgG-mediated patching/cross-linking (CL) (Methods), resulting in HA-endoglin patched and labeled by Alexa 488-GαR IgG (designated "CL: IgG αHA"), whereas myc-ALK1 is labeled by monovalent Fab′ (with secondary Alexa 546-GαM Fab′). In control experiments without HA-endoglin CL, the IgG labeling of the HA tag was replaced by exclusive Fab′ labeling. Where indicated, inhibin A (4 nM) was added during the fluorescent labeling step and maintained throughout the measurement. Representative FRAP curves are depicted in panels (i–iii), showing the lateral diffusion of singly expressed myc-ALK1 (i), singly expressed HA-endoglin immobilized by IgG CL (ii) and of myc-ALK1 in the presence of co-expressed and IgG-crosslinked HA-endoglin in the presence of inhibin A (iii). Panels (iv–v) depict average $R_f$ (iv) and $D$ values (v) of multiple experiments. Bars represent mean ± SEM values, with the number of measurements (each conducted on a different cell) shown in each bar. Some of these numbers are lower in the $D$ values panels, since only $R_f$ can be extracted from FRAP curves yielding less than 20% recovery. Asterisks indicate significant differences between the $R_f$ values of the pairs indicated by brackets (****$p < 1 \times 10^{-15}$; ***$p = 1 \times 10^{-9}$; one-way ANOVA followed by Bonferroni post-hoc test). **d** Cells were labeled for patch/FRAP using protocol 2 (Methods), leading to immobilization (CL) of the myc-endoglin and Fab′ labeling of HA-ALK4, whose lateral diffusion was then measured by FRAP. (i) Average $R_f$ values. (ii) Average $D$ values. Bars are mean ± SEM with number of measurements ($n$) depicted in each bar. Asterisks indicate significant differences between the $R_f$ values of the pairs indicated by brackets (****$p < 1 \times 10^{-15}$; **$p = 5.6 \times 10^{-3}$; one-way ANOVA followed by Bonferroni post-hoc test). No significant differences were found between $D$ values following myc-endoglin immobilization.

rhodamine-dextran was present at 5.5 times higher levels in shControl tumors compared to shINHA tumors indicating higher vascular permeability within the tumors in the presence of inhibin (Fig. 7cii–iii). To further characterize the differences in the vasculature between shControl and shINHA tumors, blood vessels were stained with CD-31 to evaluate vessel number and size (Fig. 7d). We find an increase in the total number of blood vessels in shControl tumors compared to shINHA tumors (Fig. 7di, iii). Quantitation of the size of the vessels revealed significantly smaller vessels in shControl tumors as compared to the shINHA tumors (Fig. 7dii, iii). These data together demonstrate that reducing inhibin in the tumor decreases vascular leakiness, alters vessel size and numbers and promotes more normalized vasculature in the tumors.

## Discussion

Hypoxia significantly impacts several aspects of tumor progression by regulating pathways that can be targeted for cancer management, particularly angiogenic mechanisms. We have identified inhibins', that are biomarkers for ovarian and other cancers and a member of the TGFβ superfamily, to be targets of the hypoxic response. We significantly extended our previous findings[19] to demonstrate that hypoxia induced tumor growth, angiogenesis and vascular leakiness is accompanied with, and dependent on inhibin levels in cells and tumors, and relevant to the ovarian cancer patient population. In keeping with this, hypoxia induced tumor growth can be suppressed by treatment with a selective inhibin antibody that leads to a shift in the angiogenic balance in tumors. We also provide mechanistic evidence for the involvement of ALK1 and CD105/endoglin in inhibin's effects on permeability via increased VE-cadherin internalization. Due to the lack of systemic inhibin expression in post-menopausal women, establishing the therapeutic significance of targeting inhibin in this patient population may be particularly beneficial to evade systemic side effects seen with targeting other hypoxia associated angiogenic pathways.

Significant information exists on the cycling levels of inhibins' in premenopausal women, the decline of inhibin during peri-menopause, and as a marker whose decline defines the onset of menopause leading to complete absence of inhibin in normal post-menopausal women[13]. Contrastingly, several studies have reported elevated levels of Inhibin in a subset of cancers[14,15,17,23]. Our studies shed light on the potential mechanisms leading to elevated inhibin. We also find that total inhibin is elevated in the ascites fluid of patients with ovarian cancer, a hypoxic environment that aids in dissemination of shed ovarian cancer spheroids[22,30] (Fig. 2). Serum inhibin and CA125 levels are both markers for ovarian cancer[17,18] and were also positively correlated with each other in these patient ascites (Supplementary Fig. 8a). Menopause status was unknown in these patients however the median age of the cohort was 62 and only two patients were below 50 years of age (Supplementary Fig. 8b). INHA expression and hypoxia are also correlated through a hypoxia gene score (Fig. 2d). Supporting our hypothesis that inhibin is regulated by hypoxia, we also found that exposure of ovarian cancer cells to hypoxia increased INHA expression and inhibin secretion most significantly at 0.2% oxygen after 24 h (Fig. 1) While a trend towards increase in INHA was seen at higher oxygen tensions (2.5% and 1%), these did not reach significance and could be due to additional time or additional factors needed at higher oxygen levels. Surprisingly, we did not note consistent and statistically significant increases in the activin subunits, INHBA or INHBB which only appeared to be moderately elevated (Supplementary Fig. 1b) indicating that increased inhibin secretion levels were driven by inhibinα. Previous reports indicate activin, specifically INHBA, increases in response to hypoxia in endothelial cells[57]. However, here the increase in INHA levels in endothelial cells in response to hypoxia was only moderate as compared to in tumor cells (Supplementary Fig. 1c) suggesting a potential competition in the tumor microenvironment between activin and inhibin. The inhibin ELISA used here does not detect dimeric or free activin, as it is specific to inhibinα and detects all forms of inhibin, namely inhibin A/B and free alpha subunit. Hence, a hypoxia dependent increase in the alpha subunit (INHA), in the absence of a change in activin mRNA levels (Supplementary Fig. 1b), could potentially shift the dimerization of the beta subunits (INHBA/INHBB) from activin homodimers to inhibin heterodimers.

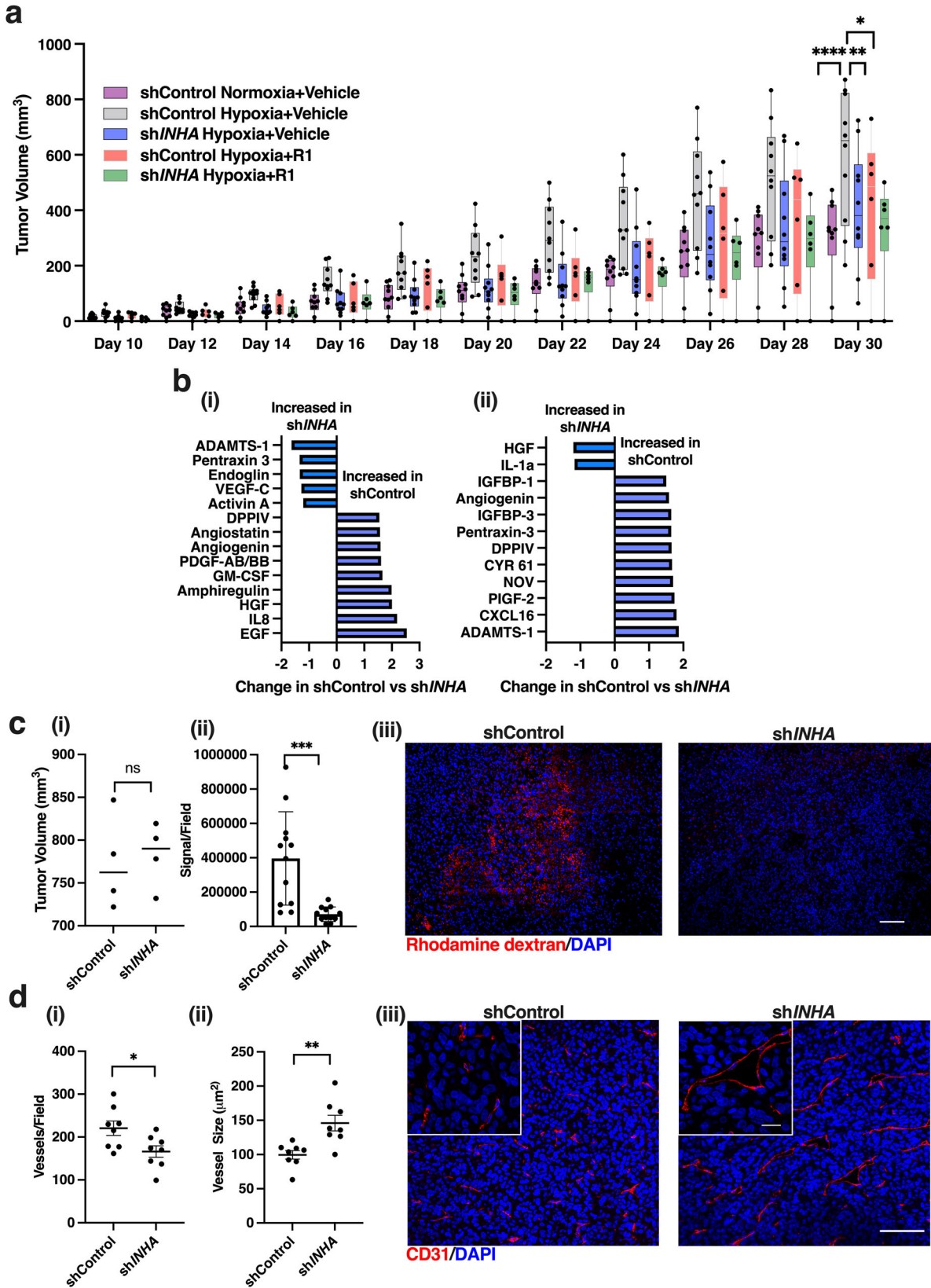

Evidence of HIF-1 dependency was observed when hypoxia exposed cells were re-exposed to oxygen (reoxygenation). HIF-1 levels returned to near baseline levels after 1 h which corresponded with *INHA* expression decreasing to levels not significantly different than baseline normoxic levels (Fig. 1dii). We did observe elevated, although non-significant, *INHA* in OV90

cells after 1 h of reoxygenation which could be attributed to mRNA turnover mechanisms. Mechanistically, we find through knockdown studies and ChIP studies that *INHA* expression is likely regulated through the HIF-1 transcription factor binding directly to the *INHA* promoter (Fig. 3). Our findings on a hypoxia response in a pathological condition as seen here, is consistent

**Fig. 7 Hypoxia regulated inhibin promotes tumor growth and regulates vascular normalization in vivo. a** Growth curves of subcutaneously implanted HEY shControl or shINHA tumors exposed to either normoxia or hypoxia (0.2% $O_2$) 24 h prior to injection. 10 mg/kg R1 antibody or vehicle control was intraperitoneally injected three times a week. Data shown as box plots where center line is median, box limits are upper and lower quartile, $n = 10$ for vehicle and $n = 6$ for R1 receiving groups. **$p < 0.01$; ****$p < 0.0001$, Two-way ANOVA followed by Tukey's multiple comparison test. **b** Fold change of proteins most altered in shControl and shINHA tumors (**a**) using the (i) human or (ii) mouse angiogenesis proteome array. ($n = 2$ tumors per group). **c** (i) Average tumor volume of HEY shControl or shINHA subcutaneous tumors used for analysis of vasculature and permeability in ii and iii. Mean ± SEM, $n = 4$. (ii) Quantitation of extravasated rhodamine-dextran (red) shown as signal per 10x field from tumors in Fig. 7ci (Methods). Mean ± SD. $n = 12$ fields from 4 tumors. ***$p < 0.001$, unpaired $t$-test. (iii) Representative images of rhodamine-dextran (red) extravasation into either shControl or shINHA subcutaneous tumors from c.i Scale bar:100 μm. **d** (i–ii) Quantitation of average (i) vessel number and (ii) size in a 10x field using ImageJ (Methods). Mean ± SD. $n = 8$ which represents averages of 8 fields in four tumors from c.i. (iii) Representative images of CD-31 (red) staining in HEY shControl and shINHA subcutaneous tumors from Fig. 7ci. Scale bar:100 μm, insets scale bar: 20 μm. *$p < 0.05$; **$p < 0.01$, unpaired $t$-test.

with a previous report demonstrating that FSH can drive *INHA* expression in granulosa cells dependent on HIF-1 in what appeared to be in an indirect manner[58]. Intriguingly, evidence for *INHA* regulation by hypoxia, specifically dependent on HIF isoforms, has been demonstrated in cytotrophoblasts[59]. Here we present detailed evidence of regulation by HIF-1, with HIF-1 interacting at *INHA*'s promoter under hypoxia (Fig. 3). cAMP and PKA can be activated in response to hypoxia as well. However, the PKA inhibitor, H89, was not able to reduce hypoxia induced *INHA* expression indicating that cAMP may not be involved in the hypoxia transcriptional regulation of *INHA* (Supplementary Fig. 3d). This does not preclude a role for cAMP-PKA in the regulation of *INHA* as it is well-established that the cAMP-PKA signaling axis enhances tumorigenesis in ovarian cancer[60]. As the effect of forskolin was additive on *INHA* expression, cAMP and PKA could represent an alternative or additive mechanism of regulation of *INHA* in ovarian cancer.

In prostate and adrenocortical cancers reports of both increased and decreased inhibinα levels have been reported[15,61,62]. In adrenocortical tumors with lower *INHA* levels, methylation of the *INHA* promoter was reported to occur at the CpG island within the proximal HRE site that we identified, suggesting potential roles for epigenetic regulation of *INHA* as well[62]. HIF transcription factors have reduced binding to methylated hypoxia response elements[63]. To this end, it is possible that not all cell lines will increase inhibin expression in response to hypoxia. If this is the case, methylation of *INHA*'s promoter may play a role making further understanding of the regulation of *INHA* expression, particularly in patients necessary in the future.

Previously we also demonstrated inhibin's effects broadly on angiogenesis[19]. Here, we sought to define more precisely the outcomes of inhibin's effects on angiogenesis, specifically in the context of hypoxia. Using recombinant inhibin and antibodies to the alpha subunit of inhibin, we find novel roles for inhibin as a permeability inducing factor with implications for tumor cell extravasation (Fig. 4). Inhibin induced permeability was dependent on ALK1 and endoglin (Fig. 6a). The VE-cadherin dependent mechanism of permeability observed by us (Fig. 5) is consistent with prior findings on the effects of other TGFβ family members' roles in promoting vascular permeability, specifically BMP6[64]. BMP6 induced vascular permeability was mediated through the Type 1 receptor ALK2[64], whereas we expect the Type 1 receptor ALK1 to be more critical for inhibin induced vascular permeability. Interestingly, inhibin strongly increased the stable interaction between ALK1 and endoglin (Fig. 6c), in line with our observation that both endoglin and ALK1 are required for permeability, and endoglin being critical for VE-cadherin internalization (Fig. 6). These findings have broad implications for other TGFβ family members that may regulate permeability dependent on Type 1 receptors. The patch/FRAP studies (Fig. 6) support our current and previous findings[19]. Although there are

some reports suggesting that inhibin can bind to ALK4[50], our findings show that inhibin does not enhance endoglin-ALK4 complex formation but rather weakens it (Fig. 6d). We have previously demonstrated that endothelial cells such as HMEC-1 express very little ALK4 compared to ALK1[19], supporting the idea that inhibin acts in endothelial cells preferentially via ALK1 in line with a potential physiological relevance of inhibin-mediated increase in endoglin-ALK1 interactions. However, these findings in endothelial cells do not contradict the current understanding of inhibin's function in non-endothelial cells, which may express more ALK4 than ALK1. These findings also do not allow us to conclude whether the ALK1-endoglin complex, which is enhanced by the binding of inhibin, is signaling or kinase competent, as non-signaling receptor complexes may exist and impact signaling in an indirect manner. Such complexes were previously reported in the context of activin and ALK2[65,66] and need further examination for inhibins.

Targeting inhibin through shRNA knockdown and antibody treatment was found to be an effective anti-angiogenic strategy leading to reduced vascular permeability increased blood vessel size but fewer number of vessels and a likely more normalized vasculature (Fig. 7d). Interestingly, in our analysis of the angiogenic proteome of HEY tumors, permeability promoting cytokines, EGF, IL-8, and DPP4 were significantly lower in shINHA tumors which were less permeable as compared to shControl tumors (Fig. 7b). Interestingly, the shINHA tumor cells produced more activin and endoglin compared to shControl tumors (Fig. 7bi). Increased activin fits the profile of the shINHA tumors expressing more anti-angiogenic proteins as activin has been shown to inhibit angiogenesis[7] which could also be a result of decreased inhibinα leading to a shift in the balance to increased dimerization of INHBA/B and thereby activin levels. Similarly, increases in tumor cell endoglin levels in shINHA tumors in vivo may reflect compensatory responses to changes in inhibin expression consistent with recent reports on endoglin expression changes in ovarian cancers[67]. Whether these changes impact metastasis and angiogenesis and are directly related to changes in inhibin levels in patients remains to be examined. Which of these altered proteins contributes the most to the either pro or anti-angiogenic tumor microenvironment remains to be determined as we unravel new roles for inhibins. In the mouse host cells where inhibin is likely to interact with endoglin from the endothelia to affect angiogenesis, endoglin levels were slightly higher in shControl receiving hosts that had more vessels compared to shINHA (Fig. 7dii). These findings also suggest that blocking inhibin could shift the balance between pro and anti-angiogenic genes.

We also demonstrate that anti-inhibin in a therapeutic regimen can reduce tumor growth in vivo (Fig. 7a). The subcutaneous model utilized does not induce ascites formation, unlike the intraperitoneal model used previously, where mice with shINHA tumors produced less ascites than those with shControl tumors[19].

However, this model was chosen as it better allows for evaluation of the vasculature in vivo and short exposure to hypoxia leads to increased tumor growth as seen here (Fig. 7a) and as seen in other models as well[68,69]. Our findings that inhibin is elevated in patient ascites (Fig. 2b) supports the idea that inhibin may promote ascites formation, likely through increased vascular permeability. The effectiveness of anti-angiogenic therapies is attributed to increased vascular normalization resulting in reduced intra-tumoral hypoxia, perfused and functional vessels that improve delivery of other chemotherapeutics and enhanced immune response[70]. Further studies utilizing intraperitoneal or intrabursal models that present additional steps of disease progression and metastasis are warranted to evaluate hypoxia and anti- inhibin approaches therein. Resistance to current anti-angiogenic therapies is also common and inhibin A levels have been reported to be increased in patients non-responsive to anti-angiogenic therapy[71] (combination of TRC105 and Bevacizumab) indicating inhibin as a potential alternative mechanism of angiogenesis in tumors resistant to other anti-angiogenic therapies. Further studies exploring the impact of anti-inhibin therapy on the effectiveness of chemotherapeutics and anti-tumor immune response as well is most certainly warranted.

In conclusion, our study shows that targeting inhibin is an effective anti-angiogenic strategy. We demonstrate a contextual mechanism for the regulation of inhibin directly driven by hypoxia and HIF-1 and fully define inhibin's contributions to hypoxia induced angiogenesis. Based on our findings and the previously known physiological functions of inhibin, we speculate that targeting inhibin may have potential improved therapeutic value in post-menopausal cancers including a significant percentage of ovarian cancers.

## Materials and methods

**Cell lines and reagents**. Ovarian epithelial carcinoma cell lines were obtained as described in key resources table in Supplementary Data 2 and were from ATCC, the NCI cell line repository through an MTA, or were as indicated. Cell line authentication was performed at the Heflin Center for Genomic Science Core Laboratories at UAB. HMEC-1s were grown per ATCC instructions. COS7 cells were grown in Dublecco's modified Eagle's medium with 10% FBS, 100 U penicillin/streptomycin and L-glutamine. Mouse embryonic endothelial cells (MEEC) WT and ENG−/− were grown as previously described[47]. Epithelial carcinoma cell lines HEY, OVCA420, SKOV3 and PA1 were cultured in RPMI-1640 containing L-glutamine, 10% FBS and 100 U of penicillin–streptomycin[72]. OVCAR5 and HEK293 were cultured in DMEM containing 10% FBS and 100 U of penicillin–streptomycin. ID8ip2Luc was a kind gift from Jill Slack-Davis[73] and cultured in DMEM containing 4% FBS, 100 U of penicillin–streptomycin, 5 µg/mL of insulin, 5 µg/mL of transferrin, and 5 ng/mL of sodium selenite. All cell lines were maintained at 37 °C in a humidified incubator at 5% $CO_2$, routinely checked for myco-plasma and experiments were conducted within 3–6 passages depending on the cell line. For hypoxia experiments, a ProOx model C21 was used and set to 0.2% $O_2$ and 5% $CO_2$. Anti-inhibin PO/23 and R1 antibodies were obtained from Oxford-Brookes university through an MTA and from Biocare Medical. INHA promoter driven luciferase reporter construct was generated through restriction cloning into pGL4.10 luciferase plasmid. Primers were designed to 547 base pairs of the INHA promoter containing the first HRE site with Nhe1 and Xho1 restriction sites on the ends. Insert was amplified from PA1 genomic DNA. Insert was ligated into pGL4.10 plasmid with T4 DNA ligase and INHA promoter region was verified through Sanger sequencing. Additional details on resource is provided in Supplementary Data 2.

**Generation of cell lines**. INHA and ARNT knockdown were generated in HEY cells infected with shRNA lentivirus, followed by selection in 2.5 µg/ml Puromycin and stable cell lines maintained in 1 µg/ml Puromycin. Luc/GFP cell lines were generated using pHIV-Luc-ZsGreen construct. Transient DNA transfections in HEK293 were performed using Lipofectamine 3000. siRNA transfections were performed using RNAiMax. In HEK293 transfections, single siRNA was used while pooled siRNA was used in HEY and OV90 transfections. Lentiviral particles were generated at the Center for Targeted Therapeutics Core Facility at the University of South Carolina. shRNA and siRNA sequences are listed in Table 1.

**RNA isolation and RT-qPCR**. Total RNA was harvested using Trizol/Chloroform extraction. RNA was transcribed using iScript Reverse Transcription Supermix and

**Table 1 shRNA and siRNA sequences.**

| Target | Source | Identifier | Target Sequence |
|---|---|---|---|
| shINHA | Sigma Aldrich | TRCN0000063904 | CCTCGGATGGAGGTTACTCTT |
| shARNT | Sigma Aldrich | TRCN0000003818 | CATTGTCCAGAGGGCTATTAA |
| siHIF-1 | Dharmacon | J-004018-07-0002 | GAACAAAUACAUGGGAUUA |
| siHIF-1 | Dharmacon | J-004018-08-0002 | AGAAUGAAGUGUACCCUA |
| siHIF-2 | Dharmacon | J-004814-06-0002 | GGCAGCACCUCACAUUUGA |
| siHIF-2 | Dharmacon | J-004814-07-0002 | GAGCGCAAAUGUACCCAAU |

**Table 2 qRT-PCR primer sequences.**

| Primers | Forward | Reverse |
|---|---|---|
| *Human* | | |
| RPL13A | AGATGGCGGAGGTGCAG | GGCCCAGCAGTACCTGTTTA |
| INHA | CGCTCAACTCCCCTGATGTC | GGGTACACGATCCACCGTTC |
| VEGF | CGCCAACCACAACATGCAG | GCTCCACGAAGGATGCCAC |
| ENG | GCCATCCAATCGAGACCCTG | TGATTGTTGGACTCCTCAGTG |
| TGFBR3 | CGTCAGGAGGCACACACTTA | CACATTTGACAGACAGGGCAAT |
| INHBA | GAACGGGTATGTGGAGATAGAG | TGTTCCTGACTCGGCAAA |
| INHBB | GCGCGTTTCCGAAATCATCA | AGGTTCTGGTTGCCTTCGTT |
| ARNT | TGACTCCTGTTTTGAACCAGC | CTGCTCACGAAGTTTATCCACAT |
| *Mouse* | | |
| RPL13A | CAAGGTTGTTCGGCTGAAGC | GCTGTCACTGCCTGGTACTT |
| INHA | AGGAAGATGTCTCCCAGGCT | GTTGGGATGGCCGGAATACA |
| VEGF | ACGACAGAAGGAGAGCAGAAG | ATGTCCACCAGGGTCTCAATC |
| *ChIP Primers* | | |
| INHA HRE1 | GGGATGTTCAGGTCCATCAG | CACACTGTAGTTGTGCAGTCAA |
| INHA HRE 2 | CCTCGTTCACCCAGAAGGTC | GATTCCGGCGTCTACGTGTG |

iTaq Universal SYBR Green Supermix. Expression data was normalized to RPL13A. qRT-PCR primer sequences are listed in resource Table 2.

**ELISA**. Inhibin ELISA's were performed according to the manufacturer's instructions for the quantitative measurement specifically of total inhibin protein (does not detect activin), that detects inhibin A (dimer of INHA/INHBA), inhibin B (dimer of INHA/INHBB), and free inhibin alpha subunit (INHA), from conditioned media of tumor cells. Cells were grown to 80% confluency in 24 well plates before media was replaced with fresh full serum media. Cells were placed in hypoxia chamber for 24 h and media was collected and concentrated using Amicon Ultra centrifugal filter.

**In vitro permeability assay**. In vitro permeability assay was adapted from Martins-Greene[74]. $1 \times 10^5$ HMEC-1 cells were plated onto a Matrigel coated 3 µM trans-well filter in full serum media. After 24 h, a second layer of $1 \times 10^5$ HMEC-1 was plated on top to obtain a confluent monolayer of cells. After an additional 24 h, media was replaced with serum free media in the top of the trans-well and either conditioned media (with 2 µg of either R1, PO23, or IgG) or serum free media containing growth factor in the bottom chamber as indicated in legends. FITC-dextran was added to the lower chamber (10 µg/ml). At indicated time points 10 µL aliquots were taken from the top chamber in triplicate and measured using microplate reader for FITC-dextran passage. At end point, filters were stained with crystal violet to confirm equal monolayers were achieved.

**Trans-well migration assay**. 75,000 HMEC-1 were plated on a fibronectin coated (10 µg/mL) 8 µm trans-well filter in serum free media. Conditioned media (with 2 µg of either R1, PO/23, or IgG) or serum free media containing 1 nM inhibin A or VEGFA was used as a chemoattractant in the bottom chamber. After 24 h, unmigrated cells were scraped off the apical side, migrated cells were fixed in methanol:acetic acid, and nuclei were stained with Hoechst. Three random images were taken per filter using 10X objective on EVOS M7000 microscope. Nuclei were counted using ImageJ.

**Trans-endothelial migration assay**. HMEC-1 were grown on 8 µm trans-well filters as per permeability assay. HMEC-1 monolayer was treated with 1 nM inhibin A or untreated for 4 h. After 4 h of treatment, 150 000 HEY-LucGFP expressing cells were plated on top of the HMEC-1 monolayer and allowed to invade for 18 h. Filters were fixed in 4% paraformaldehyde, cells on the apical side of the filter were scraped off, and filters were mounted on glass slides for imaging. Migration of GFP+ cells was visualized using 10x objective on EVOS M7000 microscope. Three random fields were captured per filter and GFP+ cells were counted using ImageJ software. Thresholding, circularity and size gating were used to exclude unmigrated cells and artifacts.

**Chromatin immunoprecipitation**. Chromatin immunoprecipitation protocol was adapted from ABCAM. Briefly, OV90 or OVCAR5 cells were grown in 150 cm$^2$ dishes until 80% confluency was reached. Cells were kept under normoxia or placed in the hypoxia chamber set at 0.2% $O_2$ for either 12 h (OVCAR5) or 24 h

(OV90). DNA was crosslinked using 0.75% formaldehyde and sheared by sonication to fragment sizes between 100–400 bp. DNA was immunoprecipitated with Dyna-beads and either HIF-1α antibody or Normal Rabbit IgG as a control. DNA was purified using Purelink PCR Purification kit and amplified using RT-qPCR with ChIP primers.

**Luciferase assay**. HEK293 cells were seeded into 24 well plate and co-transfected with a luciferase reporter containing 547 base pairs of the *INHA* promoter (pGL4.10 *INHA*) and a SV-40 (Renilla internal control vector). For HIF-1 over-expression, cells were also co-transfected with pcDNA3-HA-HIF1aP402A/P564A or PCDNA3.1. One day after transfection, cells were left in a normoxia incubator or moved to hypoxia chamber (0.2% $O_2$) for 24 h. Luciferase activity was measured using the Dual Luciferase Reporter Assay System by calculating the ratio between luciferase and Renilla and normalized to normoxia or PCDNA3.1 as indicated in legends.

**Immunofluorescence**. HMEC-1 cells were grown to confluence on fibronectin (10 μg/mL) and treated with either 1 nM inhibin A or VEGFA for 30 min in serum free media. Cells were fixed in 4% paraformaldehyde and permeabilized with 0.2% TritonX-100, followed by blocking with 5% BSA in PBS for 1 hr. VE-cadherin was labeled with anti-VE-cadherin antibody overnight at 4 °C followed by AlexaFluor 488 secondary antibody. F-actin was stained with rhodamine-phalloidin and nuclei were labeled with DAPI. Immunofluorescence imaging was performed on EVOS M7000 microscope or Nikon A1 confocal microscope. Actin fibers were quantified by measuring anisotropy using the FibrilTool Plugin in ImageJ[75].

**VE-cadherin internalization**. HMEC-1 cells grown to confluence on fibronectin (10 μg/mL) coated glass coverslips. Cell surface VE-cadherin was labeled with anti-VE-cadherin antibody at 4 °C for 30 min, washed with ice-cold PBS, and incubated at 37 °C for 30 min with 1 nM inhibin A, 1 nM VEGFA, or serum free media. After internalization was stimulated with growth factor at 37 °C, anti-VE-cadherin antibody on the cell surface was removed with mild acid wash. Internalized VE-cadherin was visualized by immunofluorescence microscopy. Internalized VE-cadherin was quantified using BlobFinder software[64,76]. Nuclei and cytoplasm were delineated and the number of signals per cell was used to quantify internalized VE-cadherin fluorescence.

**Cell surface biotinylation**. Briefly, MEEC WT or /ENG−/− were grown to confluence on gelatin coated dishes. Cell surface proteins were labeled with 2 mg/mL Sulfo-NH-SS biotin for 30 min at 4 °C. After labeling, cells were treated with 1 nM inhibin A or untreated in serum free media for 30 min at 37 °C or left at 4 °C for cell surface control samples. After treatment, cell surface biotin was removed with 20 mM MESNA buffer and internalized biotin labeled protein was isolated with neutravidin resin. Internalized biotin labeled VE-cadherin was detected by Western Blot.

**Epitope-tagged plasmids and transfection of COS7 cells for patch/FRAP studies**. The following plasmids were donated by Prof. G. C. Blobe, Duke University Medical Center: HA-tagged endoglin (endoglin-L) in pDisplay, myc-endoglin generated by PCR incorporation of the myc tag sequence into untagged endoglin in pDisplay and re-cloned in pcDNA3.1, and HA- or myc-tagged ALK1 in pcDNA3.1[77]. Human ALK4 with C-terminal myc-DDK tags in pCMV6 was obtained from OriGene Technologies (Rockville, MD), and subcloned into pcDNA3.1 by PCR followed by restriction digest and re-ligation. A stop codon was introduced at nucleotide 1516 to delete the C-terminal tags to generate untagged ALK4. This was followed by insertion of N-terminal HA tag by overlapping PCR after nucleotide 72 to generate extracellularly tagged HA-ALK4. All constructs were verified by sequencing. COS7 cells were transfected using TransIT-LT1 Mir2300 according to manufacturer's instructions. For Patch/FRAP experiments, cells grown on glass coverslips in 6-wells plates were transfected with different combinations of these vectors encoding myc- and/or HA-tagged receptor constructs. The amounts of the vectors (between 0.5 and 1 μg DNA) were adjusted to yield similar cell surface expression levels, determined by quantitative immunofluorescence.

**Fluorescent antibody labeling and IgG-mediated cross-linking for patch/FRAP**. COS7 cells were transfected with various combinations of the above epitope-tagged expression vectors. After 24 h, The cells were serum-starved (1% FBS, 30 min, 37 °C), washed with cold Hank's balanced salt solution (HBSS containing 20 mM HEPES, pH 7.2) and 2% BSA (HBSS/HEPES/BSA), and blocked with normal goat γ-globulin (200 μg/ml, 30 min, 4 °C). For FRAP studies on singly expressed receptors, the cells were then labeled successively at 4 °C in HBSS/HEPES/BSA (45 min incubations) with: (i) monovalent murine Fab′ anti myc tag (αmyc) or anti HA tag (αHA; 40 μg/ml), prepared from the respective IgGs as described by us earlier[78]; (ii) Alexa 546-Fab′ goat anti mouse (GαM; 40 μg/ml), prepared from the respective F(ab')₂ as described[79]. For patch/FRAP studies, they were labeled by one of two protocols. Protocol 1 employed successive labeling with: (i) monovalent mouse Fab′ αmyc (40 μg/ml), alone or together with HA.11 rabbit

αHA IgG (20 μg/ml) and (ii) Alexa 546-Fab′ GαM (40 μg/ml) alone or together with Alexa 488-IgG goat anti rabbit (GαR; 20 μg/ml). This protocol results in the HA-tagged receptor crosslinked and immobilized by IgGs, whereas the myc-tagged receptor, whose lateral diffusion is then measured by FRAP, is labeled exclusively by monovalent Fab′. Alternatively, we employed protocol 2 for immobilizing the myc-tagged receptor and measuring the lateral diffusion of a co-expressed Fab′-labeled HA-tagged receptor: (i) monovalent mouse Fab′ αHA (40 μg/ml) together with chicken IgY αmyc (20 μg/ml) and (ii) Cy3-Fab′ donkey anti mouse (DαM; 40 μg/ml) together with FITC-IgG donkey anti chicken (DαC; 20 μg/ml). In experiments with inhibin A, the ligand was added after starvation along with the normal goat γ-globulin and maintained at the same concentration throughout the labeling steps and FRAP measurements.

**FRAP and patch/FRAP**. COS7 cells co-expressing epitope-tagged receptors labeled fluorescently by anti-tag Fab′ fragments as described above were subjected to FRAP or patch/FRAP experiments as described[49]. FRAP studies were conducted at 15 °C, replacing samples after 20 min to minimize internalization. An argon-ion laser beam (Innova 70 C, Coherent, Santa Clara, CA) was focused through a fluorescence microscope (Axioimager.D1; Carl Zeiss MicroImaging, Jena, Germany) to a Gaussian spot of $0.77 \pm 0.03$ μm (Planapochromat 63x/1.4 NA oil-immersion objective). After a brief measurement at monitoring intensity (528.7 nm, 1 μW), a 5 mW pulse (20 ms) bleached 60–75% of the fluorescence in the illuminated region, and fluorescence recovery was followed by the monitoring beam. Values of $D$ and $R_f$ were extracted from the FRAP curves by nonlinear regression analysis, fitting to a lateral diffusion process[49]. Patch/FRAP studies were conducted analogously, except that IgG-mediated cross-linking of epitope-tagged endoglin preceded the measurement[49].

**Patient ascites**. Specimens from patients diagnosed with primary ovarian cancer was collected and banked after informed consent at Duke University Medical Center, with approval for the study from Duke University's institutional research ethics board. ELISA's were conducted using ELISA for Total inhibin from Ansh labs (#AL-134).

**Public data mining**. Clinical data and normalized RNA-seq were obtained from cBioportal[32]. The ovarian serous cystadenocarcinoma (TCGA, PanCancer Atlas) and breast invasive carcinoma (TCGA, PanCancer Atlas) were assessed for *INHA* expression and hypoxia (Buffa or Winter) scores. *INHA* expression was plotted against hypoxia score for each patient for correlation analysis.

**In vivo assays**. All animal studies and mouse procedures were conducted in accordance with ethical procedures after approval by UAB's IACUC prior to study commencement.

**Matrigel plug assay**. Matrigel plugs were formed using 200 μL of Matrigel mixed with 50 μL of HEY conditioned media and injected subcutaneously into the underside of BALB/c female mice aged 5–6 weeks. For conditioned media, HEY cells were grown until 80% confluence in 24 well plate before media was replaced with fresh full serum media. Cells were placed in hypoxia chamber for 24 h and media was collected and concentrated to 50 μL Savant SpeedVac SPD1030. Conditioned media was incubated with 2 μg of either R1 or IgG overnight before injection. Plugs were harvested 12 days after injection and hemoglobin content was determined according to Drabkin's method[19].

**In vivo subcutaneous tumor growth and permeability analysis**. $3 \times 10^6$ HEY cells either exposed to normoxia or hypoxia (0.2% $O_2$) for 24 h were subcutaneously injected into right flank of 6-week old Ncr Nude mice (Taconic). Tumor volume $((L \times W^2)/2)$ was calculated by caliper measurements every other day starting at day 10 until harvest at day 30. In animals receiving anti-inhibin treatment, R1 (BioCare) was administered IP at 2 mg/kg three times weekly. Da Vinci Green diluent (BioCare) was administered as vehicle.

For measurement of permeability, tumors were harvested between 700–800 mm³. At end point, Rhodamine dextran 70 000 MW was intravenously injected at 2 mg/kg 2 h before euthanasia. Tumors were fixed in 10% NBF and sections were analyzed for rhodamine dextran by immunofluorescence on EVOS M7000. Three sections per tumor were quantified and four images per section were taken. Thresholding was performed in ImageJ and kept constant for all images. ROUT analysis ($Q = 10\%$) was performed to test for outliers.

For tumor hypoxia analysis, tumors were harvested at varying sizes between 200–1400 mm³. Pimonidazole (HydroxyProbe) was injected intravenously at 60 mg/kg 1 h before sacrifice. Tumors were fixed in 10% NBF and sections were analyzed for pimonidazole adducts using anti-pimonidazole monoclonal antibody.

**Immunofluorescence on tissues**. Briefly, formalin fixed, paraffin-embedded tissues from subcutaneous tumors were deparaffinized by sequential washing with xylene, 100% ethanol, 90% ethanol, 70% ethanol and distilled water for 10 min each. Antigen retrieval was performed by boiling tissues in sodium citrate buffer

(pH 6.0). Blocking was performed with Background Punisher. Primary antibodies, anti-pimonidazole (1:50) and anti-CD-31(1:100), were diluted in Da Vinci Green Diluent and incubated overnight at 4 °C in a humidified chamber followed by AlexaFluor 594 secondary antibody. Nuclei were stained with DAPI. 10x images were acquired on EVOS M7000 microscope.

Quantitation of CD-31 labeled vessel size and number as well as pimonidazole was performed in ImageJ. Images were converted to binary and thresholding mask was applied equally to all images. For CD-31, objects smaller than 25 pixels were removed as were deemed too small to be vessels. For each image, average vessel size (area) and average vessel number was measured. Four images per section and two sections per tumor were used for quantitation. For pimonidazole, a 10x stitched image comprising the whole tumor section was used. The total area covered by signal was acquired and divided by total tumor area to calculate the % hypoxic area for each tumor.

**Angiogenesis proteome array**. Angiogenesis proteome array was performed according to manufacturer's instruction (R&D Systems, Supplementary Data 2). Briefly, tissues were homogenized in PBS with 1% TritonX-100 and PI cocktail. 200 µg of protein was used per sample (two samples for shControl and shINHA tumors each). Pixel intensity was quantified for each dot using ImageStudio software after background subtraction.

**Statistics and reproducibility**. All data are representative of three independent experiments, unless otherwise described in legends. Statistical analyses were performed using GraphPad Prism 9, with statistical test chosen based on experimental set up and specifically described in the figure legends. Data are expressed as mean ± SEM. Difference between two groups was assessed using a two-tailed $t$-test. Multiple group comparisons were carried by the analysis of variance (ANOVA) using One or Two-way ANOVA followed by appropriate post-hoc tests as indicated in Figure legends.

**Ethics approval and consent to participate**. Specimens from patients diagnosed with primary ovarian cancer were collected and banked after informed consent at Duke University Medical Center, with approval for the study from Duke University's institutional research ethics board. All animal studies and mouse procedures were conducted in accordance with ethical procedures after approval by UAB's IACUC prior to study commencement.

**Reporting summary**. Further information on research design is available in the Nature Research Reporting Summary linked to this article.

## Data availability
All data generated or analyzed during this study are included in this published article and its Supplementary Information files. Uncropped blots can be found in Supplementary Figs. 9–14. The underlying source data for all graphs and charts can be found in Supplementary Data 1. Additional relevant data is available upon reasonable request.

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

## Acknowledgements
We would like to acknowledge Mehri Monavarian, Victoria Alers and Hailey Young for technical assistance. We also thank the UAB High Resolution Imaging facility (HRIF), the Functional Genomics Core at University of South Carolina for lentiviral preparations and the UAB Heflin Center for genomic science core laboratories at UAB for cell line authentication services. Funding for this work was provided by NIH R01CA219495 to Mythreye Karthikeyan (KM). The funders had no role in study design, data collection and analysis, decision to publish, or preparation of the manuscript.

## Author contributions
B.H., S.P., R.C., E.L., A.S.C., M.S., L.Q.-M., and Y.L. conducted experiments. B.H. and K.M. conceptualized, designed, analyzed data, wrote and edited the manuscript. K.M. provided funding for the studies. Y.H. designed, analyzed and wrote the manuscript. A.B.N., N.H. and N.Y.L. edited the manuscript and analyzed data. R.W. and A.B. provided clinical samples.

## Competing interests
The authors declare no competing interest.
