## [Peer Review File · Communications Biology]

Reviewers' comments:

Reviewer #1 (Remarks to the Author):

In this paper "Hypoxia regulated inhibin promotes tumor growth and vascular permeability by ACVRL1 and CD105 dependent VE-cadherin internalization", Horst and colleagues investigated regulation of inhibin under hypoxia and how hypoxia-induced inhibin promoted ovarian tumor development. They concluded that inhibin was regulated specifically through HIF-1, shifting the balance from activins to inhibins. Hypoxia upregulated inhibin to promote tumor growth and endothelial cell invasion. Inhibin facilitated permeability of endothelial cells by increasing VE-cadherin internalization via ACVRL1 and CD105. Using knockdown and anti-inhibin strategies to target inhibin, inhibin-induced tumor growth and endothelial cell invasion and permeability can be reduced in vivo.

Although this work provides some interesting observations, the conclusion needs more data to back up.

Key issues:

Why use 0.2% O₂ for hypoxia condition? Oxygen levels in most of the hypoxic tumor tissues are around 1-2%. What is the physiological oxygen level in OC?

Fig. 1: The correlation between HIF1a and inhibin is not consistent across cell lines. In HEY cells, HIF1a level was relatively high under normoxia. Why the level of inhibin in these cells was not higher than that in OV90? Also, take PA1 and OVCAR5 for example, upon hypoxia, HIF1 level in PA1 was significantly elevated to a higher level than OVCAR5. For the classic HIF1 target VEGF, the induction followed the HIF1 level. PA1 showed a higher VEGF increase than OVCAR5. However, this phenomenon was not observed in Inhibin. If inhibin induction under hypoxia is exclusively regulated by HIF1 (as the authors described in the manuscript), one should expect to see similar trend between induction of VEGF and inhibin upon HIF1 stabilization. The authors should discuss this inconsistency in the manuscript. The authors should also consider to perform double staining of VEGF and inhibin.

Fig. 2: To demonstrate that HIF1-inhibin regulation is a common mechanism in OC cells, the authors should use at least two OC cell lines for each experiment and describe why those cell lines are used. Why used HEK-293 cells but not the OC cells to determine the impacts of HIF1 vs. HIF2 on inhibin expression (Fig. 2C)? Different cells respond to hypoxia differently. The authors should perform these experiments using OC cells.

What was the level of HIF1a when HIF2 was knocked down? What was the level of HIF2a when HIF1 was knockdown? To demonstrate that the knockdown is specific, the authors should perform IB to detect both HIF1 and HIF2 in Fig. 2Ci. To demonstrate that HIF1 is the dominant one to induce inhibin under hypoxia, the authors should also perform HIF1-HIF2 double knockdown and compare the inhibin level between shHIF2 and the double knockdown conditions.

Fig. 4: What is the percentage of the cells with internalized VE-cadherin upon inhibin A and VEGF A treatment?

Fig. 6: It is really surprising to see that exposure to hypoxia only for 24 hours was able to make the tumor grow faster and bigger over 30 days. Did the authors check how sensitive inhibin expression and secretion are to oxygen level? After exposure to 24 hours, if the cells are moved to normoxia condition, how long would it take for the inhibin to go back to the basal level? Since the authors showed that inhibin is regulated by HIF1a, and HIF1a is sensitive to oxygen level, if HIF1a expression reduces after cells are released from hypoxic stress, inhibin should also reduce. What is the oxygen level in the subcutaneous area? How about intra bursa injection? Will the similar results observed in the ovary sac?

Other issues:

Fig. 1: Why ID8 was not used in VEGF detection? Is there any specific reason to pick OV90 and

Hey to perform experiments in Fig. 1B and 1C? Why not use all the cell lines in Fig. 1A for the rest of the experiments?

Supp Fig. 1: Since different cell lines may have different response time to hypoxic stress, why only 24h time point was used to detect INHA in HMEC-1 cell?

Supp Fig. 3: Why suddenly switched to ID8 to test whether cAMP is involved in INHA regulation? How about other human OC cells?

Supp Fig 4: what are those two black lines on top of bands of 5 and 15 min?

Supp Fig. 7D: what does % of normoxia mean?

Reviewer #2 (Remarks to the Author):

This paper aim to study the link between inhibin and vascular permeability and tumor growth in ovarian cancer.

The paper is generally well written with clear experimental design, results presentation and discussion.

Nevertheless I would recommend some attentions/modifications to try to improve paper's quality.

1. Paper title seems to me too specific, with precise results (such as ACVRL1, CD105). I would recommend a more generic title such as : "Inhibin's contributions to hypoxia induced angiogenesis and tumor growth in ovarian cancer"...

2. Line 76. "inhibin alpha PROTEIN levels".

3. Line 125. INHalpha luciferase construct and not INHA. You speak about inhibit alpha promoter and not inhibin A protein.

4. You define hypoxic condition as 0.2% oxygen that you compare to "normoxic" conditions at 21%. But what are the real physiologic conditions ? Maybe 0.2% in tumor center, but certainly not 21% in normal tissue; even if it is the usual "normal" conditions in tissue or cell culture. Please provide a graph showing the INHA expression regarding the oxygen %. You will find a linear relation or a cut-off value ?

5. In figure 1B, C you mixed results from protein expression with ELISA and mRNA expression. Your experiment do not support your assertion "While INHA was increased three to five-times in response to hypoxia (Fig. 1A), INHBA and INHBB levels were unchanged in the two cell lines evaluated (Fig. 1C), indicating that changes in inhibin protein levels (Fig. 1B) were largely related to increases in inhibina." since you study mRNA BA and BB. Please use specific elisa with two antibodies (one related to alpha chain and the other one to beta chain), or western blot.

6. Significance of inhibin level variation in patient ascite is not reported in the graph; do you really show that "...mL in the ascites fluid with increasing concentrations found in higher stages of disease (Fig. 1E). "?

7. In figure 2C ii you want to show that the effect is driven through HIF-1 and not HIF-2. Nevertheless, using siRNA technology, you do not suppress the INHalpha expression in hypoxic conditions... It could be due to the siRNA technology itself, but... Your conclusions would be stronger if you use another technology such as cloning the inhibin promoter, and compare the hypoxia effect between the native promoter with mutation/deletion of both HRE, ...

Thank you

Editor (Remarks to the Author):

The concerns relating to the hypoxia condition and the generalization of the data across ovarian cancer cell line used in this study should be addressed. In addition, the concerns regarding the inhibin expression should be addressed.

We thank the editor for highlighting the key aspects. We have addressed all these central issues experimentally by providing new experiments on a range of oxygen tensions and gradients and reoxygenation studies (New Figs. 1A, 1Biii, 1D; Fig. 3C, Supplementary Fig. 1A and B, Supplementary Fig. 3Dii, and Supplementary Fig. 4B), data in additional cell lines (Fig. 3C and Supplementary Fig. Dii) and inhibin expression related to HIF with additional HIF knockdown experiments (New Fig. 3 and 1). In addition to these, every question raised by both reviewers have also been addressed. Please see our point-by-point responses.

Reviewer #1 (Remarks to the Author):

In this paper “Hypoxia regulated inhibin promotes tumor growth and vascular permeability by ACVRL1 and CD105 dependent VE-cadherin internalization”, Horst and colleagues investigated regulation of inhibin under hypoxia and how hypoxia-induced inhibin promoted ovarian tumor development. They concluded that inhibin was regulated specifically through HIF-1, shifting the balance from activins to inhibins. Hypoxia upregulated inhibin to promote tumor growth and endothelial cell invasion. Inhibin facilitated permeability of endothelial cells by increasing VE-cadherin internalization via ACVRL1 and CD105. Using knockdown and anti-inhibin strategies to target inhibin, inhibin-induced tumor growth and endothelial cell invasion and permeability can be reduced in vivo.

Although this work provides some interesting observations, the conclusion needs more data to back up.

Key issues

1. Why use 0.2% O₂ for hypoxia condition? Oxygen levels in most of the hypoxic tumor tissues are around 1-2%. What is the physiological oxygen level in OC?

This is a challenging issue to address in ovarian cancer and only few studies exist with inferred evidence to this end. In ovarian cancer patients using EF-5/PET scans to detect intratumor hypoxia (a compound very similar to pimonidazole which detects O₂ concentrations below 10mmHg or below 1%, PMID: 21251211) it was found that 46% of patients had hypoxia staining indicating that tumors were experiencing concentrations below 1% (PMID:32910291). In other gynecological cancers such as cervical, median O₂ concentrations are below 1% (PMID: 9288845 and **12758238**). Solid tumors experience a large range of oxygen concentrations including anoxic regions as the reviewer is likely fully aware (PMID: 33374581, 26105538). Considering the broad range of oxygen tensions and the reviewers' comments, we have now provided **New Fig. 1A** where we examine a range of O₂ tensions (also see point #9). We find that *INH*A expression is elevated in cell lines most reproducibly and significantly at 0.2% O₂ with some increase seen at 1% as well (lines 335-44).

2. Fig. 1: The correlation between HIF1α and *INHA* is not consistent across cell lines. In HEY cells, HIF1a level was relatively high under normoxia. Why the level of inhibin in these cells was not higher than that in OV90? Also, take PA1 and OVCAR5 for example, upon hypoxia, HIF1 level in PA1 was significantly elevated to a higher level than OVCAR5. For the classic HIF1 target VEGF, the induction followed the HIF1 level. PA1 showed a higher VEGF increased than OVCAR5. However, this phenomenon was not observed in Inhibin.

In our original experiment western blots for HIF-1 were not run side by side for the different cell lines (hence were shown separately in the prior version), rendering side by side assessments unreliable. We have now resolved this and provided a single western blot with samples run side-by-side from all the human cell lines used. Relatively equal HIF-1 levels can now be seen in **New Fig. 1Biii**.

3. If inhibin induction under hypoxia is exclusively regulated by HIF1 (as the authors described in the manuscript), one should expect to see similar trend between induction of VEGF and inhibin upon HIF1 stabilization. The authors should discuss this inconsistency in the manuscript. The authors should also consider performing double staining of VEGF and inhibin.

We have strengthened our primary finding on HIF1 regulation of *INHA* in **New Fig. 3C** where we have provided new evidence with two additional cell lines (ovarian) using siRNA to HIF-1, siRNA to HIF-2 and the combination of siRNA to HIF-1 and HIF-2 that demonstrate more robustly the contribution of HIF-1 (lines 440-452).

As regards the reviewer's comments on co-localization of the two proteins, this is an interesting one. Given that the current manuscript is focused on *INHA* regulation by hypoxia with VEGF used as a control, their interactions, colocalizations etc will be examined in a future study.

4. Fig. 2: To demonstrate that HIF1-inhibin regulation is a common mechanism in OC cells, the authors should use at least two OC cell lines for each experiment and describe why those cell lines are used.

We have now included 2 additional OC cell lines for siHIF-1 and 2 knockdown and added in the combination of the two siRNAs, that demonstrate that HIF-1 is the likely driver of *INHA* expression. (**New Fig. 3Ci, ii**). We have also utilized two pooled siRNAs in these experiments (see also point #3) instead of the single siRNA that was used for the HEK293 in the original experiment.

5. Why used HEK-293 cells but not the OC cells to determine the impacts of HIF1 vs. HIF2 on inhibin expression (Fig. 2C)? Different cells respond to hypoxia differently. The authors should perform these experiments using OC cells. What was the level of HIF1a when HIF2 was knocked down? What was the level of HIF2a when HIF1 was knockdown? To demonstrate that the knockdown is specific, the authors should perform IB to detect both HIF1 and HIF2 in Fig. 2Ci.

Same as for point #3 and point #4: we have now used two additional OC cell lines in **New Fig. 3Ci, ii** with a combination of the siRNAs also added. As described in the text, HEK293 was chosen due to ease of their use in luciferase assays (multiple plasmid transfections) and the relatively equal expression of HIF-1 and HIF-2. We have now also demonstrated specificity of knockdown in these HEK293 cells and moved this data to **New Supplementary Fig. 3C**.

6. To demonstrate that HIF1 is the dominant one to induce inhibin under hypoxia, the authors should also perform HIF1-HIF2 double knockdown and compare the inhibin level between shHIF2 and the double knockdown conditions.

Same response as points #3-5

7. Fig. 4: What is the percentage of the cells with internalized VE-cadherin upon inhibin A and VEGF A treatment?

These analyses have now been included in New **Supplementary Fig. 4B** showing that the percentage of cells with internalized VE-cadherin upon inhibin A treatment is 54% and 42% for VEGF A treated cells (lines 582-85).

8. Fig. 6: It is really surprising to see that exposure to hypoxia only for 24 hours was able to make the tumor grow faster and bigger over 30 days.

Several prior studies have made such observations where a similar increase in tumor volume of ovarian cancer xenografts implanted subcutaneously after short pre-exposure to hypoxia, making our observations not unique on this point (PMID: 28878214, 21320311). We have now included these citations in our discussion as well (line 832-34).

9. Did the authors check how sensitive inhibin expression and secretion are to oxygen level?

We thank the reviewer for this. We have now expanded our findings to include a new detailed analysis of changes to *INHA* gene expression levels at various oxygen tensions of 0.2%, 1%, 2.5%, 5%, 10%, and 21% in two OC cells lines and have added this as part of **New Fig. 1A** and find that *INHA* is most increased at 0.2% O₂. (also see point #1).

10. After exposure to 24 hours, if the cells are moved to normoxia condition, how long would it take for the inhibin to go back to the basal level? Since the authors showed that inhibin is regulated by HIF1a, and HIF1a is sensitive to oxygen level, if HIF1a expression reduces after cells are release from hypoxic stress, inhibin should also reduce to basal levels after re-oxygenation.

To address this question, we conducted a detailed time course analysis of HIF-1 α level changes after reoxygenation. We find that HIF-1 α levels begin to drop within 5 mins of re-oxygenation and return to baseline by 60 minutes (in two cell lines). These data are now in **New Fig. 1Di**. We then performed a time course in the same cells to determine changes in *INHA* expression upon reoxygenation. We find that *INHA* expression in HEY cells dropped to near normoxia levels after 1hr of re-oxygenation. Slightly elevated levels remained up to 18 hrs in both cell lines but did not reach statistical significance. A similar outcome was observed in the second cell line as well (lines 375-85). These findings are now in **New Fig. 1**. and are consistent with HIF-1 dependency and robustly supported by siRNA to HIF-1 as well (see point # 3-5).

11. What is the oxygen level in the subcutaneous area? How about intra bursa injection? Will the similar results observed in the ovary sac?

We performed pimonidazole staining on the subcutaneous tumors in **Fig. 2Ci and Supplementary Fig. 2A** which were slightly larger than those in Figure 6. Pimonidazole staining, detects oxygen concentrations below 10mmHg (~1.2%). Even in the smaller tumors (<500mm³) there was pimonidazole staining suggesting that the tumor cells are experiencing oxygen tension below (1.2%). Prior studies using subcutaneous tumors of similar size using an ovarian cell line demonstrated a mean O₂ concentration of 0.25% O₂. (PMID: 19623660). We have described better the sensitivity of pimonidazole staining in the text on lines 406-08. We could not find reports of oxygen measurements in the bursal sac or tumors implanted there, but such other sites can be investigated in the future and have been discussed in lines 839-41.

Other issues:

12. Fig. 1: Why ID8 was not used in VEGF detection? Is there any specific reason to pick OV90 and Hey to perform experiments in Fig. 1B and 1C? Why not use all the cell lines in Fig. 1A for the rest of the experiments?

We have now included *VEGFA* expression for ID8ip2 in **Supplementary Fig. 1A**. HEY and OV90 are high grade serous (HGS) lines that represent the predominant ovarian cancer subtype. Hence these were used for endothelial cells assays and ELISAs.

13. Supp Fig. 1: Since different cell lines may have different response time to hypoxic stress, why only 24h time point was used to detect INHA in HMEC-1 cell?

We have now included a 12hr time point for *INHA* expression in HMEC-1 in **New Supplementary Fig. 1C** showing no significant increases at this time point as well (lines 370-73).

14. Supp Fig. 3: Why suddenly switched to ID8 to test whether cAMP is involved in INHA regulation? How about other human OC cells?

We have now performed this experiment in a second human cell line (OV90) and included in **New Supplementary Fig. 4Dii**. H89 did not have an effect on hypoxia induced *INHA* expression in any of the cell lines tested (lines 497-99).

15. Supp Fig 4: what are those two black lines on top of bands of 5 and 15 min?

They are artifacts on the membrane that occurred during scanning that cannot be removed digitally.

16. Supp Fig. 7D: what does % of normoxia mean?

The growth rate of the cells was normalized to levels when under normoxia and are shown as percent growth relative to normoxia. The y-axis has been edited to clarify that and states % growth change relative to normoxia.

Reviewer #2 (Remarks to the Author):

This paper aim to study the link between inhibin and vascular permeability and tumor growth in ovarian cancer. The paper is generally well written with clear experimental design, results presentation and discussion. Nevertheless, I would recommend some attentions/modifications to try to improve paper's quality.

1. Paper title seems to me too specific, with precise results (such as ACVRL1, CD105). I would recommend a more generic title such as: "Inhibin's contributions to hypoxia induced angiogenesis and tumor growth in ovarian cancer"...

We have changed the paper title to be broader and less precise.

2. Line 76. "inhibin alpha PROTEIN levels".

Thank you, this has been corrected.

3. Line 125. INHalpha luciferase construct and not INHA. You speak about inhibit alpha promoter and not inhibin A protein.

We have clarified the wording in this sentence to indicate that we are utilizing a *INHA* promoter driven luciferase reporter construct both in methods and everywhere else utilized (lines 123, 472).

4. You define hypoxic condition as 0.2% oxygen that you compare to "normoxic" conditions at 21%. But what are the real physiologic conditions? Maybe 0.2% in tumor center, but certainly not 21% in normal tissue; even if it is the usual "normal" conditions

in tissue or cell culture. Please provide a graph showing the INHA expression regarding the oxygen %. You will find a linear relation or a cut-off value?

We appreciate this comment that 21% does not reflect the actual oxygen tension in normal tissue therefore as suggested, we have now included a New Fig. 1A where we examine a range of O₂ tensions including two lower oxygen concentrations 10% and 5% to portray the range of oxygen concentrations more accurately in the tissue. We find no statistically significant differences between the ~21% seen in the normal incubator and other more physiologically relevant concentrations such as 10% or 5%. We find that *INHA* expression is elevated in cell lines most reproducibly and significantly at 0.2% O₂ with some increase seen at 1% as well (lines 335-44). Also see point #1.

5. In figure 1B, C you mixed results from protein expression with ELISA and mRNA expression. Your experiment do not support your assertion "While INHA was increased three to five-times in response to hypoxia (Fig. 1A), INHBA and INHBB levels were unchanged in the two cell lines evaluated (Fig. 1C), indicating that changes in inhibin protein levels (Fig. 1B) were largely related to increases in inhibin α ." since you study mRNA BA and BB. Please use specific elisa with two antibodies (one related to alpha chain and the other one to beta chain), or western blot.

We regret the confusion. *INHA* encodes for Inhibin alpha that as a protein can exist either as free inhibin alpha or dimerized to a beta subunit (product of either *INHBA* or *INHBB*). The ELISA is a clinical grade ELISA and highly specific to inhibin alpha (INHA) regardless of whether it is free or dimerized. It does not detect free *INHBB* or free *INHBA* or dimeric forms of *INHBA/INHBB* (activin A/B respectively). Hence our ELISA findings are consistent with inhibin alpha mRNA changes seen throughout the manuscript. We have now edited the activin language and added additional clarification sentences in methods (lines 146-49), results and adjusted the language in discussion (lines 750-54).

6. Significance of inhibin level variation in patient ascites is not reported in the graph; do you really show that "...mL in the ascites fluid with increasing concentrations found in higher stages of disease (Fig. 1E). "?

We have changed the language to state what the levels are, as ascites is mostly present in advanced stage disease with limited ascites in patients with lower stage disease (lines 401-03).

7. In figure 2C ii you want to show that the effect is driven through HIF-1 and not HIF-2. Nevertheless, using siRNA technology, you do not suppress the INHalpha expression in hypoxic conditions... It could be due to the siRNA technology itself, but... Your conclusions would be stronger if you use another technology such as cloning the inhibin promoter, and compare the hypoxia effect between the native promoter with mutation/deletion of both HRE, ...

We appreciate that in the HEK293 the siHIF-1 knockdown did not result in a full suppression of *INHA* under hypoxia. We have now added two additional human OC cell lines to perform the siHIF-1/2 knockdowns, used pooled siRNAs, and have also utilized a combination knockdown that together demonstrate that HIF-1 is the dominant driver of *INHA* expression under hypoxia (**New Fig. 3Cii-iii**). These findings along with the new reoxygenation experiments (**New Fig. 1D**) firmly demonstrate HIF-1 as the main regulator. Also please see Reviewer 1 points #3-6 comments with attached figures.

REVIEWERS' COMMENTS:

Reviewer #1 (Remarks to the Author):

The authors answered all my concerns. I have no further comment.

Reviewer #2 (Remarks to the Author):

Thank you for the improvements of the paper.

Nevertheless I still have some comments about INHA expression through HIF-1.

1. In your new figure 1 you clearly show the presence of HIF-1 at oxygen concentration of 2.5 and 1%, but without increase of INHA expression, but with a stimulation of VEGF expression. How could you explain this difference in HIF effect ?

I suspect that 24h exposition to hypoxia is too short to see an effect on INHA expression

2. In your answer to my comment #7 about the INHA expression you still use knockdown experiment to support the role of HIF-1 and not HIF-2.

Once again, your conclusions would be stronger if you use another technology such as co transfection with your luciferase reporter plasmid containing the INHA promoter and plasmids allowing overexpression of HIF1A and HIF 2.

Indeed, it could be possible that you do not see an effect of HIF2 down regulation because there is no HIF 2 at 24h hypoxia. We could hypothesize that HIF 1 is present in short term hypoxia and HIF2 in long term hypoxia.

Reviewer #2 (Remarks to the Author):

Reviewer #2 (Remarks to the Author):

Thank you for the improvements of the paper.

Nevertheless I still have some comments about INHA expression through HIF-1.

1. In your new figure 1 you clearly show the presence of HIF-1 at oxygen concentration of 2.5 and 1%, but without increase of INHA expression, but with a stimulation of VEGF expression. How could you explain this difference in HIF effect? I suspect that 24h exposition to hypoxia is too short to see an effect on INHA expression

This is a possibility and we have included in discussion that at higher oxygen tensions longer time or additional factors may be needed (line # 516-519 in manuscript article word file)

2. In your answer to my comment #7 about the INHA expression you still use knockdown experiment to support the role of HIF-1 and not HIF-2.

Once again, your conclusions would be stronger if you use another technology such as co transfection with your luciferase reporter plasmid containing the INHA promoter and plasmids allowing overexpression of HIF1A and HIF 2.

Indeed, it could be possible that you do not see an effect of HIF2 down regulation because there is no HIF 2 at 24h hypoxia. We could hypothesize that HIF 1 is present in short term hypoxia and HIF2 in long term hypoxia

We regret not clarifying in the figures better as we included HIF1A overexpression with the luciferase reporter (Fig 3Eii) and more pertinently, in the presence of HIF2 alone (siHIF1, Fig 3C- western blot, lane 6), INHA levels were not elevated significantly (Fig 3C qRT-PCR below). However, it is possible HIF2 may play a role in non-cancer/ other systems that can be explored in other studies.